# Improving Generalization and Safety of Deep Neural Networks with Masked Anchoring

## Abstract

Anchoring is a recent architecture and task-agnostic technique that can produce state-of-the-art epistemic uncertainty estimates, and improve extrapolation capabilities. However, the differences between anchored models and non-anchored variants is not well studied – as there is little insight into the kinds of functions anchoring induces and how they behave under distribution shifts. In this paper, we analyze and improve anchoring as a training protocol for deep neural networks, evaluating them on important tasks of out of distribution generalization, task adaptation, anomaly rejection and calibration. We pinpoint the impact of anchoring on generalization as being inversely related to the sensitivity of the model to the distribution of residuals. We further improve this sensitivity using a new technique called Random Anchor Masking (RAM) that significantly improves the quality of anchored models. We build evidence for the superiority of RAM-training using a range of benchmarks of varying size, using neural networks of varying complexity and scale.

## 1 Introduction

Anchoring is a simple, architecture-agnostic protocol for training neural networks; it has enabled several capabilities ranging from state-of-the-art uncertainty estimates and calibration (Thiagarajan et al., 2022), outlier rejection (Anirudh & Thiagarajan, 2022), and extrapolation (Netanyahu et al., 2023). At a high level, anchoring replaces the input to the network using a tuple comprising an "anchor" randomly chosen from the training dataset, and the residual between the input and the anchor. This is done such that, the prediction on the input should be consistent regardless of the anchor choice. By reposing the prediction task into the joint space of anchors and residuals, this trivial transformation has been shown to provide significant gains in performance over standard deep models.

However, the behaviour of anchored training is not sufficiently clear from the existing literature. For example, in (Thiagarajan et al., 2022), the fact that anchoring leads to meaningful uncertainties is justified via studying anchoring as perturbations to the neural tangent kernel (NTK) (Jacot et al., 2018); while this explains why meaningful uncertainties arise, it does not shed light into the quality of functions anchoring can produce. Moreover, in (Netanyahu et al., 2023), the anchored model is essentially considered equivalent to the non-anchored model in terms of the function it approximates. As a result, there is little clarity on anchoring as a training mechanism on its own, and its impact on generalization and safety characteristics.

This is essential as models are being increasingly adopted in a variety of applications across different domains such as healthcare (Davenport & Kalakota, 2019) and autonomous driving (Bogdoll et al., 2022), where it is prudent to holistically evaluate and understand the model behavior under challenging distribution or task shifts. In that context, generalization to data beyond the training distribution (Yang et al., 2021) as well as the ability to accurately detect changes in the input data are critical to promote safe model adoption (Hendrycks et al., 2021b). While generalization includes producing accurate predictions on covariate shifts *i.e.*, images of a particular modality collected from different sensors or adapting to to new task shifts (Andreassen et al., 2021), sensitivity to input data variations include producing well-calibrated confidences (Guo et al., 2017) under data shifts and the ability to accurately detect anomalous samples (Hendrycks & Gimpel, 2017b) of disparate semantic characteristics with respect to the training data.

Existing research has demonstrated significant performance improvements in each task considered independently (Hendrycks & Gimpel, 2017a; Lee et al., 2018; Hendrycks et al., 2018; Sehwag et al.,

2021; Tack et al., 2020; Liu et al., 2020; Robey et al., 2021). Since a model can non-trivially trade-off between the different safety objectives, in practice, it is challenging to effectively train and assess models (Hendrycks et al., 2022). This motivates the need to devise novel training protocols that are not specific to architectures and do not require sophisticated priors (e.g, PixMix Hendrycks et al. (2022), but simultaneously improve generalization and safety of the trained models.

In this paper, we study the viability of anchoring as a training protocol for large-scale datasets and sophisticated model architectures. Using a variety of architectures of varying complexity and size, we study different aspects of generalization such as prediction under severe corruptions, calibration, anomaly rejection, adaptation under task/covariate shifts and robustness under label noise to establish the benefits of anchoring over standard network training. We summarize our key findings below:

- Anchoring is able to boost generalization performance without sacrificing on safety metrics such as calibration or anomaly rejection;

- We pinpoint the improved generalization of anchored models as being linked to the diversity of residuals exposed to the model during training;

- Building upon this insight, we propose Random Anchor Masking (RAM), an efficient and effective regularization for improving diversity, which shows significantly improved generalization over both standard anchoring and non-anchored models.

- We observe significant improvements in generalization and safety across datasets of varying size and complexity (Imagenet-1K/CIFAR100,10) and architectures of varying scales (RegNet/WRN/ResNet/WRN/DEIT-T/DEIT-S/Swinv2-T/SWINv2-S/SWINv2-B/ViT-B).

## 2 BACKGROUND AND RELATED WORK

**Notations:** Let $\mathcal{F}_\theta$ be a multi-class classifier parameterized by $\theta$ that is trained on a dataset $\mathcal{D} = \{(x_i, y_i)\}_{i=1}^M$ with $M$ samples. The classifier works on an input image $x \in X \subseteq \mathbb{R}^{C \times H \times W}$. The objective of this classifier is to predict an output label $\hat{y} \in Y$ where $Y$ is defined as the set $Y = \{1, 2, \ldots, K\}$, where $K$ represents the total number of distinct classes. Here, $x$ is an RGB image from the space of inputs $X$ with $C$ channels, height $H$, and width $W$.

**Anchoring in Deep Models:** The principle of anchoring introduced in (Thiagarajan et al., 2022) involves the reparameterization of an input $x$ into a tuple comprising an *anchor* $r$, drawn at random from an anchor distribution $P(R)$, and the *residual* $\Delta x$ denoted by $[r, \Delta x] = [r, x - r]$. For image data, the tuple is constructed by concatenating the anchor and residual along the channel axis, resulting in a $6-$channel tensor for every 3-channel RGB image $x$. Apart from this architectural change, the optimization objective of the deep network is left unchanged (Thiagarajan et al., 2022). The simple re-parameterization of the input leads to a joint distribution that depends not only on $P(R)$, but also on the distribution of the residuals $P(\Delta)$. Formally, the training objective can be written as:

$$\theta^* = \arg\min_\theta \ \mathcal{L}(y, \mathcal{F}_\theta^a([r, x - r]), \forall r \in P(R) \tag{1}$$

where $\mathcal{L}(.)$ is a loss function. Effectively, anchoring enforces that for every input sample $x$, $\mathcal{F}_\theta^a([r_1, x - r_1]) = \mathcal{F}_\theta^a([r_2, x - r_2]) = \cdots = \mathcal{F}_\theta^a([r_k, x - r_k])$, where $\mathcal{F}_\theta^a$ is the anchored model that operates on the tuple $([r_k, x - r_k])$ to predict $y$. During both training and testing, the anchors need to be drawn from $P(R)$, which is set to the training distribution $P(X)$ itself in our implementation. However, given that equation 1 explicitly marginalizes the choice of anchor, any random training sample (or a small number of them) can be used to obtain predictions at inference time. While the idea of enforcing prediction consistency across different anchor choices might appear similar to data augmentation methods, we want to clarify that anchoring does not impose any invariance to data characteristics, but only expands the space of (anchor, residual) pairs with each additional anchor. This general principle can be used with any model architecture or task, and several recent works have demonstrated the utility of anchoring in design optimization (Thiagarajan et al., 2022), reinforcement learning (Netanyahu et al., 2023), generalization gap prediction (Narayanaswamy et al., 2022) and graph neural network calibration (Trivedi et al., 2023).

**Why does anchoring improve generalization?** In order to understand this, we directly refer to the following two key results from existing literature: (a) In (Thiagarajan et al., 2022), it was shown that centering a dataset using different constant inputs (or anchors) will lead to different solutions, due to inherent lack of shift invariance in NTKs induced by commonly adopted neural networks. Building on this principle, anchored training uses different anchors for the same sample across different epochs with the goal of marginalizing out the effect of anchor choice at inference time. But in this process, it implicitly helps explore a large class of hypotheses, thus resulting in a more generalizable solution; (b) In (Netanyahu et al., 2023), it was theoretically showed that an anchored model can better extrapolate to unseen data regimes where (independently) the anchor $r \in P(R)$ and the residual for the unseen sample $\Delta x_t \in P(\Delta)$. Furthermore, it was argued that the problem of generalizing to "out of support" (OOS) samples (i.e., no evidence of observing such a sample in the training data) can be made more tractable by carefully choosing anchors $\tilde{r} \sim P(R)$ at inference time, such that $x_t - \tilde{r} = \tilde{\Delta} x_t \sim P(\Delta)$, even if the specific combination of $[\tilde{r}, x_t - \tilde{r}]$ was not observed during training. While such an out-of-combination (OOC) setting can still be challenging to handle, the hope is that the predictions might be better calibrated, *i.e.*, low confidence for OOC tuples.

Building upon this insight, we argue that, by exposing the model to more diverse combinations of (anchor, residual) pairs during training, generalization can be systematically improved. To this end, we explore a novel regularization strategy to effectively increase the diversity of $P(\Delta)$. Furthermore, existing works have not rigorously studied the viability of anchoring as a training protocol for large-scale datasets, modern architectures or even practical distribution shifts. Hence, for the first time, we empirically benchmark anchored training across dataset sizes (CIFAR-10 to Imagenet), architectures (ResNet to ViT) and network sizes (5M to 88M parameters), using important safety metrics including OOD generalization calibration, anomaly rejection and adaptation (both ID and OOD evaluation).

## 3 PROPOSED APPROACH

In anchoring, since $r$ is always drawn from $P(X)$ by design, handling novelty to $\Delta x_t$ becomes key to improving the OOD generalization. Intuitively, with wide anchor distributions (e.g., $P(R) = P(X)$ in ImageNet training), the residual distribution $P(\Delta)$ obtained through the extensive space of (anchor, residual) pairs is expressive enough to support a wide variety of OOD scenarios, when compared to conventional deep models. A direct implication of this statement is that an anchored model can behaves unreliably when presented with novel residuals. While this might not be of concern when test data comes from the in-distribution (ID), *i.e.*, $\Delta x \in P(\Delta)$, its effect is more pronounced when handling out of distribution (OOD) data in practical tasks such as generalizing under corruptions or distribution shifts (Shen et al., 2021), anomaly rejection (Hendrycks et al., 2019; Hendrycks & Dietterich, 2019) or adaptation under task shifts (Trivedi et al., 2023). Consequently, without ensuring sufficient generalization and safety properties, anchoring becomes a less attractive choice in practice, particularly with large-scale models.

A naïve way to improve the diversity of $P(\Delta)$ (or equivalently $P(r, \Delta)$) is to consider a much wider anchor distribution (i.e., $P(R) \supset P(X)$). However, it is non-trivial to characterize the anchor distribution (e.g., for a large dataset such as ImageNet) and to arbitrarily find additional data to improve the diversity. Even in cases where we can find such additional data, it can lead to increased computational complexity for anchored training. For example, when training on ImageNet (regardless of the architecture), we find that anchored training requires 20 additional epochs to converge to the same level of validation loss as a vanilla model. On the other hand, with CIFAR-10 or CIFAR-100, anchored models converge effectively with the standard training recipe. To circumvent these challenges, we introduce RAM (Random Anchor Masking), a simple and efficient regularization strategy that leads to improved generalization, without impacting the complexity of training.

### 3.1 RAM: IMPROVING GENERALIZATION VIA NOISY RESIDUALS

In this paper, we adopt an alternative approach to increasing diversity of $P(\Delta)$ by making the residuals noisy, and propose a simple implementation in the form of RAM. Given the inherent challenge in defining a suitable residual noise distribution (and inferring its hyper-parameters), we define the noise distribution to be same as the anchor distribution itself, i.e., $P(R) = P(N) = P(X)$, and implement it efficiently using the RAM regularizer. Formally, for a given tuple $[r, x - r]$, anchor masking zeroes out the anchor while keeping the residual fixed, i.e., $[0, x - r]$, while making the prediction for the

sample x. In general, the tuple for making a prediction for x with a zero anchor (note: zero vector is a valid anchor in our anchor distribution) should have been written as $[0, x - 0]$. In this context, anchor masking can be re-interpreted as making a prediction using the zero anchor, but with a noised residual $x + \epsilon$ where $\epsilon = -r$ and $\epsilon \in P(N)$. Using noisy residuals during training naturally improves the diversity of $P(\Delta)$ and more interestingly, avoids the over-reliance of OOD test samples only on the anchors, which can lead to highly mis-calibrated predictions.

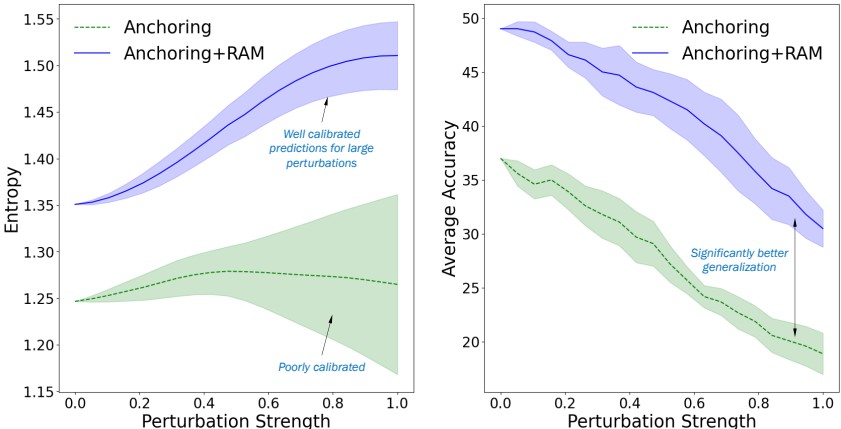

Figure 1: We examine how two anchored ResNet-18 (He et al., 2016) models trained on CIFAR10 (Krizhevsky, 2009) respond to input data corruption. We measure the entropy (left) and accuracy (right) as a function of perturbation strength. Note, the anchor is fixed to be the same in both cases. A well-calibrated model is expected to produce higher entropy predictions (less confident) as severity of perturbation increases, while the accuracy should correspondingly drop. We average these metrics across 20 perturbation strengths in $[0.0, 1.0]$, in 10 random directions applied to 100 OOD examples ($\mathrm{CIFAR} - 10\mathrm{C}/\mathrm{Gaussian\ noise}/\mathrm{severity\ 5}$). Standard deviation is measured across the 10 trials.

In addition to improving the training process, RAM controls the sensitivity of the model predictions under data noise, thus leading to significantly improved calibration in practice. Figure 1 highlights the distinction between the prediction calibration (measured in terms of the entropy) of two anchored models trained with and without RAM regularization respectively. With increasing severity of the data corruption (Gaussian noise in this case) for fixed anchor (a random sample from $P(R)$), we witness improved calibration and superior generalization behavior. This hypotheses holds true even with more challenging distribution shifts and corruptions as we demonstrate in our empirical studies. Furthermore, we find in our experiments that, RAM helps improve other safety metrics such as adaptation under task/covariate shifts and anomaly rejection. Next, we describe the implementation of anchored training with RAM regularization.

## 3.2 IMPLEMENTATION DETAILS

For all models trained in this study, we followed the training recipes from the torchvision library (https://pytorch.org/vision/stable/models.html) and directly adopted the same hyper-parameter configurations even for anchored training. The hyper-parameter $\alpha \in [0, 1]$ directly controls the schedule of residual corruption during training. Note that, with the schedule $\alpha$ ($\alpha = 0.2$ corresponds to every $5^{\mathrm{th}}$ batch), we perform anchor masking for an entire batch and obtain gradient updates with noisy residuals. However, this is only an implementation choice and one can consider alternative approaches; for example, the residual corruption can be applied to $\alpha$ fraction of samples from each batch or this can be included as an additional loss objective. While using a high $\alpha$ value can help improve generalization, it can also adversely affect the training convergence and eventually the ID performance itself. We chose $\alpha$ such that the validation loss was low on ID test data and it needs the same number of epochs as standard anchoring. Based on our experiments with CIFAR-10, CIFAR-100 and ImageNet across multiple architectures (RegNet, ResNet, WRN40-2, DEIT, ViTb and SWINv2), we recommend the use of $\alpha = 0.2$. In terms of memory overheads and inferencing efficiency, we find that anchored training is very similar to conventional neural network training

protocols. The only difference we notice is that, with larger datasets like ImageNet, anchored training requires 20 additional epochs to converge to the same level of validation loss.

# 4  EXPERIMENTS AND RESULTS

We extensively assess the performance of anchored training (w/ and w/o RAM) across diverse benchmarks, architectures and generalization tasks. Our key goal is to examine how RAM behaves across a spectrum of datasets with varying complexities and sizes, as well as different choices of model architectures. Following this, we broadly evaluate the behavior of models in three tasks: (i) Generalization to out-of-distribution (OOD) data, (ii) Anomaly rejection, and (iii) Adaptation of representations under task (and distribution) shifts.

## 4.1  SETUP

**Training Protocol:** All models unless specified are trained on ImageNet-1K (Russakovsky et al., 2015), a large-scale vision benchmark comprising of 1.3 million training images across 1000 diverse categories. Following standard practice, we adopt the optimization and pre-processing settings provided in the torchvision library to train all models. As mentioned earlier, we train all anchored models on ImageNet for an additional 20 epochs beyond the one provided in the recipe to closely match the top-1 accuracy of a non-anchored model and to effectively leverage the diversity in $P(\Delta)$.

**Architectures:** We consider a family of model architectures with different levels of structural and parameter complexity to rigorously assess the viability of anchoring as a standard training protocol. Specifically, we consider RegNetY-800-MF (6.4M) (Radosavovic et al., 2020), DEIT-T (5M), DEIT-S (22M) (Touvron et al., 2021), SWINv2-T (28.4M), SWINv2-S (49.7M), SWINv2-B (87.8M) (Liu et al., 2022) and ViT-B-16 (86.6M) (Dosovitskiy et al., 2021) models. Note that for training anchored models, we modify the first convolution layer or the transformer layer in each architecture to handle the reparameterized 6 channel images.

**Baselines and Evaluation Metrics:** We compare anchoring with RAM against the non-anchored counterparts as well as the standard anchored variants across the three tasks. We report the top-1 accuracy to evaluate model performance for OOD generalization and adaptation tasks. For assessing anomaly rejection performance, we resort to the AUROC between the ID and OOD energy scores (Liu et al., 2020). In addition, we report the calibration error on datasets considered for OOD generalization using the recently proposed Smoothed ECE metric (Błasiok & Nakkiran, 2023) to assess the quality of model confidences under different test-time conditions. Finally, we use the standard accuracy metric for evaluating adaptation fidelity.

## 4.2  MAIN FINDINGS AND DISCUSSION

**OOD Generalization:** An important assessment of model safety is the ability to generalize to distributional shifts from the training data. As such, we expect the models to encode non-trivial semantic concepts from the training data and use the same to generalize even when the test-time distributions change. To that end, we conduct a zero-shot evaluation of the pre-trained models on (i) ImageNet-C (Hendrycks & Dietterich, 2019) with 19 natural image corruptions across 5 severity levels, (ii) ImageNet-C̄ (Mintun et al., 2021) with 10 noise corruptions across 5 severity levels; (iii) ImageNet-R (Hendrycks et al., 2021a) containing different renditions of 200 classes from ImageNet; (iv) ImageNet-S (Wang et al., 2019) comprising black and white sketch images from each class of ImageNet; and (v) ImageNet-V2 (Recht et al., 2019) containing three new test datasets for ImageNet models in addition to the standard evaluation set. In Table 1, we report the OOD generalization performance across the different benchmarks. It can be observed from the results that anchored training and in particular, the variant with RAM consistently yields improvements over their non-anchored architectures for OOD generalization across all architectures and different distribution shifts. A striking observation is that, network capacity plays a significant role in effectively leveraging the increased diversity produced by RAM. For example, with ImageNet, as we move from RegNet (5M) to SWINv2-B (88M), we witness larger performance improvements over both anchoring w/o RAM as well as standard training. On the contrary with the DEIT-T model with only 5M parameters, the benefits of incorporating RAM are somewhat limited. Finally, following our observation in

Table 1: Out-of-distribution (OOD) generalization performance (corruptions and distribution shifts) on models trained with Imagenet-1K. We report the Top1 accuracy in each case and highlight the best performing model in each with pink.

| Architecture | Anchoring? | RAM? | ImageNet-v2 | ImageNet-R | ImageNet-S | ImageNet-C | | | | | ImageNet-C̃ | | | | |
|---|---|---|---|---|---|---|---|---|---|---|---|---|---|---|---|
| | | | | | | Sev. 1 | Sev. 2 | Sev. 3 | Sev. 4 | Sev. 5 | Sev. 1 | Sev. 2 | Sev. 3 | Sev. 4 | Sev. 5 |
| DEIT-T (5M) | ✗ | ✗ | 67.9 | 32.7 | 19.89 | 60.33 | 53.01 | 46.07 | 36.56 | 26.02 | 59.86 | 54.98 | 49.04 | 40.1 | 35.54 |
| | ✓ | ✗ | 68.04 | 33.01 | 20.44 | 61.01 | 53.95 | 46.89 | 37.17 | 26.51 | 60.63 | 55.77 | 49.78 | 40.98 | 36.41 |
| | ✓ | ✓ | 66.83 | 32.57 | 19.61 | 60.0 | 53.09 | 46.41 | 36.8 | 26.26 | 60.37 | 56.09 | 50.72 | 42.63 | 38.12 |
| RegNet (6.4M) | ✗ | ✗ | 71.85 | 33.03 | 22.28 | 61.31 | 51.34 | 42.8 | 31.63 | 21.02 | 56.17 | 47.99 | 40.75 | 31.73 | 26.74 |
| | ✓ | ✗ | 71.39 | 32.13 | 22.41 | 61.12 | 51.15 | 42.42 | 30.79 | 20.12 | 56.42 | 47.94 | 40.66 | 32.05 | 26.82 |
| | ✓ | ✓ | 71.43 | 32.45 | 21.51 | 62.24 | 53.46 | 45.48 | 33.89 | 22.9 | 61.47 | 54.7 | 47.46 | 38.33 | 32.75 |
| DEIT-S (22M) | ✗ | ✗ | 75.19 | 41.88 | 29.12 | 70.62 | 64.74 | 59.0 | 49.94 | 38.39 | 70.88 | 67.39 | 62.33 | 53.42 | 48.71 |
| | ✓ | ✗ | 75.88 | 43.11 | 30.01 | 71.5 | 65.81 | 60.36 | 52.05 | 41.2 | 71.78 | 68.33 | 63.19 | 54.54 | 49.87 |
| | ✓ | ✓ | 75.46 | 42.14 | 29.19 | 70.66 | 65.04 | 59.71 | 51.52 | 40.72 | 71.52 | 68.61 | 64.21 | 56.24 | 51.66 |
| SWINv2-T (29.4M) | ✗ | ✗ | 77.21 | 40.84 | 27.08 | 71.63 | 64.89 | 57.77 | 47.77 | 35.66 | 71.37 | 67.12 | 61.2 | 52.01 | 46.54 |
| | ✓ | ✗ | 77.34 | 40.36 | 27.56 | 72.32 | 65.85 | 58.95 | 49.51 | 37.41 | 72.68 | 68.96 | 63.29 | 53.74 | 48.14 |
| | ✓ | ✓ | 77.16 | 41.17 | 27.68 | 72.13 | 65.71 | 59.21 | 50.01 | 38.58 | 73.51 | 70.45 | 65.77 | 57.31 | 51.76 |
| SWINv2-S (49.7M) | ✗ | ✗ | 79.21 | 45.17 | 32.25 | 74.48 | 68.8 | 62.84 | 54.32 | 42.85 | 75.39 | 72.26 | 67.14 | 58.73 | 53.7 |
| | ✓ | ✗ | 79.3 | 45.95 | 32.08 | 74.75 | 68.87 | 63.12 | 54.7 | 43.14 | 76.07 | 73.33 | 68.79 | 60.49 | 55.19 |
| | ✓ | ✓ | 78.93 | 46.63 | 33.3 | 74.7 | 69.12 | 63.65 | 55.5 | 44.33 | 76.59 | 74.24 | 70.17 | 62.93 | 58.25 |
| VITb16 (86.6M) | ✗ | ✗ | 76.31 | 44.06 | 29.4 | 72.37 | 66.57 | 61.6 | 52.88 | 41.09 | 72.75 | 69.01 | 63.47 | 54.7 | 50.07 |
| | ✓ | ✗ | 76.17 | 45.56 | 32.32 | 72.64 | 67.14 | 62.33 | 54.46 | 43.48 | 73.21 | 69.74 | 64.57 | 56.03 | 51.46 |
| | ✓ | ✓ | 76.28 | 46.39 | 33.0 | 72.52 | 67.38 | 62.87 | 55.13 | 44.52 | 73.65 | 70.91 | 66.87 | 59.29 | 54.94 |
| SWINv2-B (87.8M) | ✗ | ✗ | 79.39 | 45.7 | 31.91 | 74.45 | 68.55 | 62.34 | 53.66 | 41.87 | 75.12 | 72.15 | 67.16 | 58.66 | 53.75 |
| | ✓ | ✗ | 79.35 | 47.6 | 33.42 | 74.95 | 69.28 | 63.43 | 55.08 | 43.8 | 76.36 | 73.3 | 68.49 | 60.05 | 54.81 |
| | ✓ | ✓ | 79.76 | 48.16 | 33.34 | 75.24 | 69.63 | 64.05 | 56.08 | 45.19 | 77.1 | 74.69 | 70.81 | 63.53 | 58.77 |

Table 2: Measuring calibration under distribution shifts and anomaly rejection performance of models trained on Imagenet-1K. We report the Smoothed ECE ($\downarrow$) and the AUROC ($\uparrow$) scores to assess calibration and anomaly rejection performance respectively. For smoothed ECE, we report the mean and standard deviation across all ImageNet OOD datasets.

| Architecture | Anchoring? | RAM? | Calibration (ECE) | Anomaly Rejection (AUROC) | | | | | |
|---|---|---|---|---|---|---|---|---|---|
| | | | | LSUN (C) | LSUN (R) | iSUN | Textures | Places365 | NINCO |
| DEIT-T | ✗ | ✗ | $0.116 \pm 0.016$ | 94.75 | 85.03 | 84.02 | 85.55 | 67.55 | 74.42 |
| | ✓ | ✗ | $0.116 \pm 0.015$ | 96.12 | 86.73 | 85.95 | 85.96 | 70.08 | 75.78 |
| | ✓ | ✓ | $0.112 \pm 0.015$ | 94.54 | 85.88 | 85.26 | 86.25 | 70.36 | 76.31 |
| RegNet | ✗ | ✗ | $0.154 \pm 0.064$ | 98.79 | 97.61 | 97.77 | 88.37 | 83.03 | 80.18 |
| | ✓ | ✗ | $0.164 \pm 0.068$ | 98.77 | 98.04 | 98.01 | 87.6 | 83.39 | 80.44 |
| | ✓ | ✓ | $0.144 \pm 0.071$ | 98.58 | 95.82 | 96.42 | 90.05 | 83.24 | 82.18 |
| DEIT-S | ✗ | ✗ | $0.11 \pm 0.029$ | 93.6 | 88.68 | 87.86 | 80.51 | 66.66 | 70.1 |
| | ✓ | ✗ | $0.112 \pm 0.027$ | 95.04 | 90.72 | 90.64 | 81.82 | 66.23 | 72.41 |
| | ✓ | ✓ | $0.113 \pm 0.027$ | 94.93 | 89.82 | 89.58 | 82.76 | 67.33 | 72.64 |
| SWINv2-T | ✗ | ✗ | $0.121 \pm 0.034$ | 91.73 | 78.93 | 80.25 | 76.83 | 72.53 | 77.46 |
| | ✓ | ✗ | $0.121 \pm 0.032$ | 89.53 | 78.73 | 78.68 | 76.64 | 74.75 | 76.49 |
| | ✓ | ✓ | $0.117 \pm 0.027$ | 90.25 | 78.15 | 77.69 | 78.09 | 77.16 | 78.49 |
| SWINv2-S | ✗ | ✗ | $0.126 \pm 0.039$ | 94.54 | 82.21 | 82.89 | 77.87 | 70.63 | 74.73 |
| | ✓ | ✗ | $0.122 \pm 0.045$ | 95.35 | 87.46 | 87.73 | 80.83 | 76.67 | 77.79 |
| | ✓ | ✓ | $0.119 \pm 0.041$ | 94.71 | 83.43 | 84.18 | 79.66 | 74.85 | 78.47 |
| VITb16 | ✗ | ✗ | $0.109 \pm 0.037$ | 91.59 | 87.34 | 86.92 | 79.24 | 65.72 | 65.98 |
| | ✓ | ✗ | $0.106 \pm 0.035$ | 90.87 | 85.81 | 85.17 | 76.88 | 66.16 | 68.49 |
| | ✓ | ✓ | $0.105 \pm 0.028$ | 89.88 | 85.5 | 84.55 | 78.91 | 67.18 | 70.32 |
| SWINv2-B | ✗ | ✗ | $0.132 \pm 0.055$ | 95.05 | 85.32 | 85.32 | 76.35 | 65.99 | 72.13 |
| | ✓ | ✗ | $0.129 \pm 0.058$ | 95.82 | 85.4 | 85.98 | 77.88 | 70.75 | 72.63 |
| | ✓ | ✓ | $0.124 \pm 0.051$ | 95.84 | 86.5 | 87.34 | 75.74 | 73.66 | 74.53 |

Figure 1, anchoring w/ RAM withstands high noise severity better than the other models, achieving improvements of $2\% - 7\%$ at severity 5.

**Calibration and Anomaly Rejection:** While generalization to distribution shifts is key to improve model utility, it must be ensured that the models produce well-calibrated prediction probabilities that match the likelihood of correctness. Hence, calibration is a vital test to understand how tempered the model predictions are under shifts and ensure that they do not provide over-confident predictions to OOD inputs. On the other hand, when the inputs are semantically disparate and do not share the same label space as the training data, we require the models to appropriately flag them as anomalies.

Table 3: LP-based adaptation for models trained on Imagenet-1K on target datasets. We measure the accuracy (↑) of the adapted model using the validation split of the target dataset.

| Architecture | Anchoring? | RAM? | DTD | UCF101 | Flowers102 | Food101 | OxfordPets | StandfordCars | CIFAR-10 | CIFAR-100 | Caltech | Aircraft | Average |
|---|---|---|---|---|---|---|---|---|---|---|---|---|---|
| DEIT (tiny) | ✗ | ✗ | 63.65 | 65.56 | 89.36 | 60.16 | 90.46 | 37.11 | 88.51 | 69.98 | 91.54 | 41.49 | 69.78 |
| | ✓ | ✗ | 63.42 | 65.0 | 89.24 | 60.44 | 89.89 | 36.28 | 89.15 | 70.82 | 91.95 | 41.19 | 69.74 |
| | ✓ | ✓ | 64.3 | 66.56 | 90.58 | 60.17 | 89.1 | 37.33 | 89.19 | 70.12 | 91.97 | 42.27 | 70.16 |
| RegNet | ✗ | ✗ | 68.09 | 70.34 | 94.03 | 66.33 | 90.43 | 53.55 | 91.08 | 73.24 | 94.0 | 54.67 | 75.58 |
| | ✓ | ✗ | 66.67 | 72.14 | 93.91 | 65.89 | 90.9 | 50.21 | 90.21 | 73.12 | 93.6 | 51.97 | 74.86 |
| | ✓ | ✓ | 68.03 | 71.21 | 94.28 | 66.24 | 90.76 | 52.87 | 92.03 | 74.74 | 93.38 | 54.64 | 75.82 |
| SWIN-v2 (tiny) | ✗ | ✗ | 72.1 | 74.28 | 95.37 | 72.5 | 92.53 | 55.76 | 91.82 | 74.89 | 94.53 | 57.49 | 78.13 |
| | ✓ | ✗ | 71.87 | 74.94 | 95.41 | 73.16 | 93.51 | 57.23 | 91.78 | 75.88 | 93.96 | 57.64 | 78.54 |
| | ✓ | ✓ | 71.34 | 74.97 | 94.88 | 72.68 | 92.34 | 54.46 | 92.12 | 75.87 | 94.58 | 58.15 | 78.14 |
| DEIT (small) | ✗ | ✗ | 68.56 | 73.57 | 93.5 | 67.17 | 91.71 | 49.61 | 92.46 | 75.89 | 94.05 | 48.96 | 75.55 |
| | ✓ | ✗ | 68.32 | 73.91 | 93.5 | 68.41 | 92.15 | 50.62 | 93.63 | 77.81 | 94.69 | 48.63 | 76.17 |
| | ✓ | ✓ | 68.5 | 74.94 | 93.46 | 68.68 | 92.37 | 50.6 | 93.41 | 77.34 | 94.81 | 49.29 | 76.34 |
| VITb16 | ✗ | ✗ | 69.21 | 76.13 | 94.97 | 71.22 | 92.59 | 59.96 | 95.58 | 81.82 | 95.22 | 56.44 | 79.31 |
| | ✓ | ✗ | 68.97 | 75.42 | 94.32 | 70.94 | 91.74 | 61.56 | 95.35 | 81.91 | 95.17 | 58.6 | 79.4 |
| | ✓ | ✓ | 68.79 | 76.47 | 94.44 | 71.25 | 92.4 | 61.95 | 96.1 | 82.07 | 94.94 | 58.54 | 79.7 |

To that end, we conduct an extensive evaluation of model calibration under distribution shifts using the ImageNet-C/C̄/R/S/V2 variants, and for anomaly rejection, we consider the benchmarks: (i) *LSUN (C)* (Yu et al., 2015), (ii) *LSUN (R)*(Yu et al., 2015), (iii) *iSUN*, (iv) *Textures* (Cimpoi et al., 2014b), and (v) *Places365* (Zhou et al., 2017) and (vi) *NINCO* (Bitterwolf et al., 2023), a recent OOD benchmark which comprises of images with semantic overlap with ImageNet but with no class overlap.

We report the calibration and the anomaly rejection performance of all models in Table 2. It can be observed that, incorporating RAM leads to significantly improved model calibration irrespective of the choice of architecture, thus demonstrating the importance of increasing $P(\Delta)$ diversity. Similar to any existing regularization strategy (e.g., Mixup) adopted for improving generalization, RAM with high $\alpha$ might have the risk of producing models with reduced sensitivity towards anomalous data (i.e., compared to anchoring w/o RAM). However, we find that with $\alpha = 0.2$, the anomaly rejection trade-off is minimal and interestingly, it can even lead to higher rejection AUROC scores on challenging cases (e.g., NINCO) using networks with higher capacity.

**Downstream Adaptation:** To investigate the effectiveness of the features obtained through the proposed training strategies, we employ two evaluation protocols: Adaptation(ID Eval) and Adaptation(OOD Eval). While both protocols involve keeping the feature extractor $\mathcal{F}_\theta$ (pre-trained on ImageNet) frozen and training a linear probe on the acquired features, the protocols primarily differ in terms of the distribution of the test set.

In Adaptation (ID Eval), we assume that the distribution of the dataset used for linear probing is the same as that of the test set, for example, performing linear probing on UCF-101 train set and evaluating on UCF-101 test samples. This protocol enables us to explore the transferability of our features under various task shifts. Through Adaptation (OOD Eval), we aim to study the transferability and generalizability of the features under both task and distribution shifts. To this end, we first train the linear probe with a dataset that introduces a task shift compared to ImageNet. We then evaluate the linear probe with data drawn from a different distribution, characterized by covariate shifts, compared to the probing dataset. Note, that we only consider covariate shifts for our evaluation.

Adaptation (ID Eval): We consider the following suite of target datasets representing varying levels of task shifts. (i) UCF101 (Soomro et al., 2012); (ii) Food101 (Bossard et al., 2014); (iii) Flowers102 (Nilsback & Zisserman, 2008); (iv) OxfordPets (Parkhi et al., 2012); (v) StanfordCars (Krause et al., 2013); (vi) DTD (Cimpoi et al., 2014a) ; (vii) Caltech101 (Fei-Fei et al., 2004); (viii) FGVC-Aircraft (Maji et al., 2013); (ix) CIFAR-10 (Krizhevsky et al., 2009); and (x) CIFAR-100 (Krizhevsky et al., 2014) datasets. We employ LogisticRegression from Scikit-Learn to derive our linear probes. The regularization parameter $C$ is determined through k-fold cross-validation.

From Table 3, we can see that, the proposed masked training of anchored models consistently yields adaptation gains on multiple datasets across different architectures. Notably, we observe an upward trend compared to non-anchored models with an increase in architecture complexity. These results

Table 4: LP-based adaptation for models trained on DomainNet on domains *Real* and *Sketch* respectively. Contrasting Table 3, we measure the zero-shot accuracy (↑) of the adapted model on the remaining domain shifted datasets in DomainNet.

| Architecture | Anchoring? | RAM? | Train: Real | | | | | Train: Sketch | | | | |
|---|---|---|---|---|---|---|---|---|---|---|---|---|
| | | | Real | Sketch | Clipart | Painting | Average | Real | Sketch | Clipart | Painting | Average |
| DEIT-T | ✗ | ✗ | 75.03 | 19.78 | 28.9 | 39.52 | 40.81 | 36.35 | 42.79 | 27.23 | 26.54 | 33.23 |
| | ✓ | ✗ | 75.1 | 20.77 | 29.45 | 39.68 | 41.25 | 36.48 | 43.58 | 28.21 | 27.17 | 33.86 |
| | ✓ | ✓ | 74.72 | 20.07 | 28.1 | 38.94 | 40.46 | 35.68 | 42.77 | 27.23 | 26.32 | 33.0 |
| RegNet | ✗ | ✗ | 79.0 | 20.78 | 32.18 | 41.4 | 43.34 | 38.73 | 47.49 | 29.87 | 27.78 | 35.97 |
| | ✓ | ✗ | 78.61 | 20.99 | 32.37 | 40.34 | 43.08 | 38.39 | 47.41 | 30.03 | 26.86 | 35.67 |
| | ✓ | ✓ | 78.52 | 20.49 | 32.04 | 41.34 | 43.1 | 38.22 | 47.48 | 29.9 | 27.97 | 35.89 |
| SWINv2-T | ✗ | ✗ | 80.88 | 22.69 | 34.91 | 44.09 | 45.64 | 44.22 | 50.3 | 33.78 | 31.91 | 40.05 |
| | ✓ | ✗ | 81.0 | 22.89 | 34.83 | 44.11 | 45.71 | 42.0 | 49.66 | 32.5 | 30.8 | 38.74 |
| | ✓ | ✓ | 80.77 | 23.09 | 35.42 | 44.3 | 45.9 | 42.23 | 49.35 | 32.61 | 30.54 | 38.68 |
| DEIT-S | ✗ | ✗ | 79.47 | 25.2 | 35.31 | 45.01 | 46.25 | 42.71 | 50.67 | 34.02 | 31.23 | 39.66 |
| | ✓ | ✗ | 79.78 | 25.76 | 36.75 | 45.22 | 46.88 | 42.3 | 51.72 | 35.28 | 31.94 | 40.31 |
| | ✓ | ✓ | 79.4 | 25.92 | 36.04 | 45.54 | 46.72 | 43.18 | 51.75 | 34.96 | 32.04 | 40.48 |
| VITb16 | ✗ | ✗ | 80.7 | 25.85 | 37.38 | 46.3 | 47.56 | 41.35 | 53.61 | 35.4 | 31.42 | 40.44 |
| | ✓ | ✗ | 80.41 | 27.67 | 39.48 | 46.66 | 48.56 | 44.55 | 55.02 | 38.24 | 33.39 | 42.8 |
| | ✓ | ✓ | 80.48 | 28.02 | 38.98 | 46.97 | 48.61 | 44.81 | 55.3 | 37.76 | 32.7 | 42.64 |

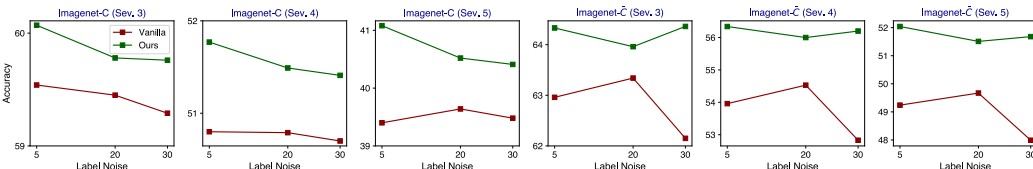

Figure 2: Evaluating OOD generalization for DeIT(S) models on Imagenet, comparing non-anchored and RAM variants trained with varying levels of label noise. Each sub-plot depicts the top-1 accuracy across 3 severities from Imagenet-C/C̄. We observe that RAM offers increased robustness to label noise compared to the non-anchored counterparts providing an accuracy improvement of ≈2%.

thus indicate that anchored training (w/ and w/o RAM) provide feature representations that are transferable even with sophisticated architectures.

Adaptation (`OOD Eval`): For this task, we use DomainNet (Peng et al., 2019), a large-scale benchmark with images from multiple domains, with images from 345 categories. More specifically, we pick four domains, namely *real*, *sketch*, *clipart*, and *painting*, to evaluate generalization of adapted probes under challenging domain shifts. We conduct two sets of out-of-domain (OOD) evaluations – one involves training the linear probe with images from the *real* domain, and the other with images from the *sketch* domain. We then directly test the linear probes on the held-out domains. From Table 4, we make similar findings as above, that anchored models produce richer representation than their non-anchored counterparts, thus leading to improved transferability. RAM regularization continues to provide improvements, and especially under complex shifts (e.g., *sketch* → *painting*), with large architectures (e.g., VITb16), we notice non-trivial performance improvements.

## 4.3 ANALYSIS

**Impact of Training Label Noise:** In the previous section, we evaluated the effectiveness of different models on generalization to OOD corruptions. While we found RAM to be particularly effective, here, we seek to further evaluate its efficacy in the realistic, but more challenging setting where the training data may be compromised for e.g., by label noise (Chen et al., 2023). Introducing label noise during training adds confusion to the system, challenging the model to maintain robustness to noise while preserving generalization. To that end, we randomly flip the labels of $l\%$ of the data during the training of a DEIT(S) model on ImageNet, and repeat this experiment with varying levels of label noise $l = \{5, 20, 30\}$. Subsequently, we evaluate OOD generalization on ImageNet C/C̄. Figure 2 illustrates that, even with higher levels of label noise and increasing data corruption complexities, anchored training w/ RAM demonstrates robustness and superior generalization compared to the non-anchored model.

Table 5: Evaluating OOD Generalization, Calibration, Anomaly Rejection, and ID Adaptation Performance for CIFAR-10/100 models trained with and without RAM. In most cases, anchored models, particularly with RAM, consistently outperform non-anchored and standard anchoring baselines in generalization, calibration, and adaptation across different architectures.

| Dataset | Architecture | Anchoring? | RAM? | Corruptions Accuracy (↑) | | | | | Calibration (↓) | Ano. Rej. (↑) | Adaptation-ID (↑) |
|---|---|---|---|---|---|---|---|---|---|---|---|
| | | | | Sev. 1 | Sev. 2 | Sev. 3 | Sev. 4 | Sev. 5 | | | |
| CIFAR-10 | ResNet-18 | ✗ | ✗ | 89.44 | 83.47 | 77.91 | 70.74 | 58.72 | 0.15 ± 0.07 | 92.46 | 19.05 |
| | | ☑ | ✗ | 88.99 | 84.28 | 79.16 | 72.09 | 59.82 | 0.14 ± 0.07 | 91.38 | 22.15 |
| | | ☑ | ☑ | 90.98 | 87.15 | 83.17 | 77.81 | 67.26 | 0.09 ± 0.05 | 94.19 | 22.61 |
| CIFAR-100 | ResNet-18 | ✗ | ✗ | 61.2 | 53.6 | 48.8 | 42.6 | 33.3 | 0.12 ± 0.04 | 75.96 | 31.61 |
| | | ☑ | ✗ | 64.5 | 55.5 | 49.9 | 43.3 | 33.1 | 0.13 ± 0.04 | 83.58 | 32.45 |
| | | ☑ | ☑ | 67.78 | 59.63 | 54.34 | 48.03 | 37.87 | 0.13 ± 0.04 | 87.48 | 34.1 |
| | WRN 40-2 | ✗ | ✗ | 62.26 | 52.82 | 46.85 | 40.12 | 30.05 | 0.26 ± 0.06 | 83.76 | 33.91 |
| | | ☑ | ✗ | 64.55 | 55.47 | 49.43 | 42.84 | 32.75 | 0.24 ± 0.06 | 84.79 | 33.64 |
| | | ☑ | ☑ | 66.0 | 57.77 | 52.33 | 45.64 | 35.52 | 0.19 ± 0.06 | 79.42 | 37.08 |

**Impact on Dataset Size:** To better understand the behavior of anchored training with smaller datasets, we conducted extensive experiments involving the training of ResNet-18 models on CIFAR-10 and CIFAR-100 datasets, along with WideResNet40-2 (WRN-40-2) on CIFAR-100, following hyper-parameters and training configurations as in (Thiagarajan et al., 2022). Similar to our ImageNet experiments, we considered OOD generalization, calibration, anomaly rejection and adaptation performance evaluations. Remarkably, our results, detailed in Table 5, highlight that the incorporation of RAM regularization during anchored training significantly enhances robustness to corruptions across all severity. The improvement in accuracy under corruptions surpasses $5.2\%$ compared to the non-anchored model and even outperforms standard anchoring by more than $4\%$.

Beyond generalization, anchoring improves calibration errors and anomaly rejection fidelity. Notably, the anchoring variant with RAM achieves the lowest calibration error on average, showcasing its effectiveness. In anomaly rejection, RAM outperforms standard anchoring by almost $3\%$ and non-anchored training by about $2\%$. Finally, linear probing on CIFAR-10 trained models provides consistent gains, with anchored models showing an average improvement of over $3.5\%$. We extended our study to WRN-40-2, on CIFAR-100 and observed the persistence of benefits in OOD generalization, calibration and adaptation. Interestingly, improvements are evident even at lower severity levels on CIFAR-100 compared to CIFAR-10. The adaptation results in Table 5 further underscore the clear advantages of RAM across both architectures. While anomaly rejection gains were substantial for ResNet-18, WRN-40-2 exhibited a trade-off between OOD generalization and anomaly rejection (discussed in Section A of the appendix).

## 5 CONCLUSION

Through this work, we find that, across varying dataset sizes (CIFAR-10 to ImageNet), model architectures (ResNet to VIT) and network sizes (5M to 88M parameters), anchored training can provide significant gains in OOD generalization, anomaly rejection and adaptation, compared to conventional training. In particular, when the train recipe includes high-capacity architectures or advanced mechanisms (e.g., Mixup, EMA, label-smoothing, cutmix), anchored training tends to provide bigger performance gains over the base models. However, we realize that state-of-the-art results in OOD generalization are often obtained using model souping (Wortsman et al., 2022) or by fine-tuning large scale pre-trained models (Goyal et al., 2023). Hence, we believe an important future direction of work is to integrate anchoring (w/ RAM) into these approaches. While we have not theoretically characterized the behavior of RAM regularization, our hypothesis is rooted in existing theory and our empirical results provide evidence for the hypothesis. However, building upon the empirical success of anchoring, carrying out a theoretical study on generalization in anchored models is crucial.

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
