# A    APPENDIX: IMPROVING GENERALIZATION AND SAFETY OF DEEP NEURAL NETWORKS WITH MASKED ANCHORING

## A.1    CHOICE OF MASKING PROBABILITIES $\alpha$ FOR CIFAR-100/ IMAGENET

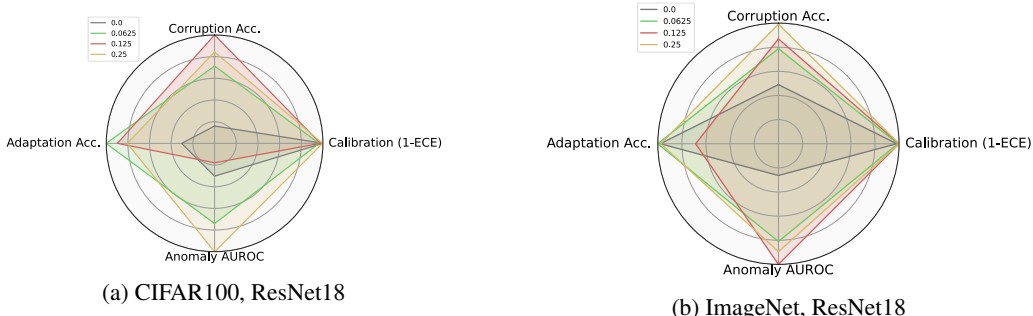

(a) CIFAR100, ResNet18

(b) ImageNet, ResNet18

Figure 3: **Impact of masking probability $\alpha$ on generalization and safety metrics**

Table 6: Anomaly Detection Performance of CIFAR-10 trained ResNet.

| Architecture | Anchoring? | Zero-crop? | $\alpha$-value | OOD Rejection (AUROC↑) | | | | | | |
|---|---|---|---|---|---|---|---|---|---|---|
| | | | | LSUN (C) | LSUN (R) | iSUN | Textures | Places365 | Tiny Imagenet | CIFAR100 |
| ResNet-18 | No | No | - | 97.4 | 94.63 | 94.05 | 87.49 | 90.65 | 95.87 | 87.16 |
| | Yes | No | - | 95.98 | 93.65 | 92.65 | 84.81 | 90.69 | 95.11 | 86.77 |
| | Yes | Yes | 0.0625 | 97.1 | 93.91 | 93.26 | 88.97 | 92.12 | 96.02 | 87.55 |
| | Yes | Yes | 0.125 | 98.19 | 96.55 | 96.12 | 91.47 | 94.59 | 97.1 | **90.14** |
| | Yes | Yes | 0.25 | 98.11 | 96.36 | 95.69 | 90.02 | 93.1 | 96.42 | 89.64 |

Table 7: Smooth Empirical Calibration Error ([Błasiok & Nakkiran, 2023](#)) of ResNet-18 trained using CIFAR-10 when evaluated on CIFAR10-C.

| Architecture | Anchoring? | Zero-crop? | $\alpha$-value | CIFAR-10C (Accuracy↑ / ECE↓) | | | | |
|---|---|---|---|---|---|---|---|---|
| | | | | Sev. 0 | Sev. 1 | Sev. 2 | Sev. 3 | Sev. 4 |
| ResNet-18 | No | No | - | 0.07 | 0.1 | 0.14 | 0.18 | 0.25 |
| | Yes | No | - | 0.06 | 0.09 | 0.13 | 0.17 | 0.24 |
| | Yes | Yes | 0.0625 | 0.06 | 0.08 | 0.11 | 0.15 | 0.21 |
| | Yes | Yes | 0.125 | 0.04 | 0.06 | 0.08 | 0.12 | 0.17 |
| | Yes | Yes | 0.25 | 0.04 | 0.06 | 0.08 | 0.11 | 0.17 |

## A.2    EXPANDED RESULTS TABLE

Tables 6 - 11 contain the individual results for the datasets CIFAR-10, CIFAR-100 and ImageNet.

# B    EXPLORING THE TRADE-OFFS BETWEEN GENERALIZATION AND SAFETY METRICS

Our findings highlight that across different datasets and architectural choices, we achieve superior generalization performance which is precisely the objective RAM was expected to achieve. However, a closer look into the other safety metrics reveals that there is a non-trivial trade-off with generalization. As an example, we can observe from Tables 1 and 2 that the AUROC for CIFAR-100 and ImageNet are respectively compromised to achieve better generalization. This naturally raises a fundamental question of identifying better methods for controlling the safety metrics. While this is an open ended question, there are two possible methods that provide us scope to better explore and understand the trade-offs.

Table 8: (Top) Anomaly detection on CIFAR-100 trained models - ResNet and WRN using a large array of benchmarks. (Bottom) Adaptation to different datasets through linear probing. We also include detailed results with varying masking probability $\alpha$.

| Architecture | Anchoring? | Zero-crop? | $\alpha$-value | OOD Rejection (AUROC ↑) | | | | | | |
| --- | --- | --- | --- | --- | --- | --- | --- | --- | --- | --- |
| | | | | LSUN (C) | LSUN (R) | iSUN | Textures | Places365 | Tiny Imagenet | CIFAR10 |
| ResNet-18 | No | No | - | 81.01 | 76.1 | 76.19 | 66.63 | 75.79 | 81.01 | 74.96 |
| | Yes | No | - | 81.36 | 85.25 | 85.73 | 83.7 | 85.56 | 85.11 | 78.37 |
| | Yes | Yes | 0.0625 | 90.34 | 86.52 | 86.67 | 83.11 | 86.07 | 90.79 | 78.68 |
| | Yes | Yes | 0.125 | 92.31 | 79.21 | 78.37 | 76.63 | 84.16 | 89.58 | 80.01 |
| | Yes | Yes | 0.25 | 89.86 | 88.58 | 87.76 | 87.87 | 88.58 | 90.29 | 79.43 |
| WRN-40-2 | No | No | - | 96.40 | 79.69 | 77.32 | 78.20 | 82.67 | 94.77 | 77.30 |
| | Yes | No | - | 97.1 | 80.38 | 79.36 | 82.58 | 81.96 | 95.76 | 76.42 |
| | Yes | Yes | 0.25 | 97.55 | 69.05 | 68.72 | 65.73 | 81.49 | 95.18 | 78.23 |

| Architecture | Anchoring? | Zero-crop? | $\alpha$-value | Adaptation (Accuracy ↑ / ECE↓) | | | | | | |
| --- | --- | --- | --- | --- | --- | --- | --- | --- | --- | --- |
| | | | | UCF101 | Food 101 | Flowers 102 | Stanford Cars | Oxford Pets | DTD | CIFAR10 |
| ResNet-18 | No | No | - | 25.27 / 0.393 | 16.67 / 0.293 | 52.54 / 0.292 | 5.61 / 0.481 | 22.29 / 0.419 | 24.65 / 0.406 | 0 / 0 |
| | Yes | No | - | 25.48 / 0.392 | 16.34 / 0.302 | 53.11 / 0.294 | 6.02 / 0.477 | 23.99 / 0.417 | 25.95 / 0.402 | 76.24 / 0.074 |
| | Yes | Yes | 0.0625 | 29.69 / 0.365 | 18.45 / 0.335 | 57.49 / 0.261 | 6.58 / 0.476 | 25.21 / 0.397 | 27.36 / 0.384 | 78.41 / 0.084 |
| | Yes | Yes | 0.125 | 29.29 / 0.369 | 18.25 / 0.409 | 55.3 / 0.271 | 6.39 / 0.477 | 26.36 / 0.405 | 27.78 / 0.382 | 77.42 / 0.086 |
| | Yes | Yes | 0.25 | 28.58 / 0.374 | 17.95 / 0.404 | 55.66 / 0.27 | 6.57 / 0.475 | 24.97 / 0.402 | 27.48 / 0.39 | 77.48 / 0.083 |
| WRN-40-2 | No | No | - | 27.20 / 0.406 | 20.80 / 0.03 | 54.61 / 0.29 | 6.75 / 0.411 | 26.63 / 0.403 | 24.47 / 0.411 | 76.92 / 0.013 |
| | Yes | No | - | 25.96 / 0.408 | 21.54 / 0.025 | 53.67 / 0.288 | 5.86 / 0.394 | 25.62 / 0.385 | 24.53 / 0.41 | 78.31 / 0.013 |
| | Yes | Yes | 0.25 | 31.75 / 0.388 | 24.11 / 0.017 | 58.99 / 0.263 | 7.29 / 0.394 | 30.23 / 0.382 | 27.01 / 0.401 | 80.19 / 0.011 |

Table 9: (Smooth) Empirical Calibration Errors of ResNet and WRN models trained on CIFAR-100 when evaluated on CIFAR-100-C.

| Architecture | Anchoring? | Zero-crop? | $\alpha$-value | CIFAR-100C (ECE↓) | | | | |
| --- | --- | --- | --- | --- | --- | --- | --- | --- |
| | | | | Sev. 1 | Sev. 2 | Sev. 3 | Sev. 4 | Sev. 5 |
| ResNet-18 | No | No | - | 0.07 | 0.1 | 0.11 | 0.14 | 0.18 |
| | Yes | No | - | 0.08 | 0.1 | 0.12 | 0.15 | 0.19 |
| | Yes | Yes | 0.0625 | 0.1 | 0.13 | 0.16 | 0.19 | 0.24 |
| | Yes | Yes | 0.125 | 0.1 | 0.13 | 0.14 | 0.17 | 0.22 |
| | Yes | Yes | 0.25 | 0.08 | 0.11 | 0.13 | 0.15 | 0.19 |
| WRN-40-2 | No | No | - | 0.19 | 0.24 | 0.26 | 0.29 | 0.34 |
| | Yes | No | - | 0.17 | 0.22 | 0.24 | 0.27 | 0.32 |
| | Yes | Yes | 0.25 | 0.12 | 0.16 | 0.19 | 0.22 | 0.27 |

One method is to control the masking probability $\alpha$ during training. Figure 4a depicts the radar plot that illustrates the relative improvements in corruption accuracy, calibration error (we present 1 - calibration error for ease of comparison), adaptation accuracy and anomaly detection over a non-anchored ResNet18 trained on CIFAR-10 across different choices of $\alpha$. It can be observed that there is direct correlation between $\alpha$ and the corruption accuracy. However, at lower $\alpha = 0.125$ there is a recovery in the anomaly detection performance with a compromise on generalization. While this method is simple to adopt, we do not have an optimal method of identifying $\alpha$, which we reserve for future work.

Since choosing $\alpha$ can be non-trivial in practice, we can fix a particular $\alpha$ and better control the training such that we regularize the residual distribution and discourage the model to solely improve generalization. As anchored models are centered upon $P(\Delta)$, we explicitly construct a multi-variate normal distribution to estimate the same while training and sample data points from the tails of the distribution. The selected tail samples are then considered as outliers and we enforce an objective in order as to maximize the entropy for those samples. Mathematically, the objective for the tail samples t is given by $\mathcal{L}_{reg}(\mathcal{U}, \mathcal{F}_\theta([0, 0 - \text{t}])$ where $\mathcal{L}_{reg}$ is the cross-entropy from the predictions to the uniform prior $\mathcal{U}$. Note that $\mathcal{L}_{reg}$ is used as regularizer with a weight $\lambda$ during training. We

Table 10: Smooth Empirical Calibration Error (Błasiok & Nakkiran, 2023) of ResNet and RegNet and ViT b-16 models trained on ImageNet evaluated using ImageNet-C an ImageNet-C̄

| Architecture | Anchoring? | Zero-crop? | $\alpha$-value | ImageNet-C (ECE↓) | | | | | ImageNet-C̄ (ECE↓) | | | | |
|---|---|---|---|---|---|---|---|---|---|---|---|---|---|
| | | | | Sev. 1 | Sev. 2 | Sev. 3 | Sev. 4 | Sev. 5 | Sev. 1 | Sev. 2 | Sev. 3 | Sev. 4 | Sev. 5 |
| ResNet-18 | No | No | - | 0.084 | 0.093 | 0.106 | 0.124 | 0.144 | 0.101 | 0.138 | 0.183 | 0.226 | 0.239 |
| | Yes | No | - | 0.083 | 0.092 | 0.109 | 0.132 | 0.152 | 0.1 | 0.141 | 0.184 | 0.223 | 0.235 |
| | Yes | Yes | 0.025 | 0.082 | 0.089 | 0.098 | 0.114 | 0.13 | 0.096 | 0.134 | 0.174 | 0.211 | 0.224 |
| | Yes | Yes | 0.05 | 0.081 | 0.087 | 0.096 | 0.111 | 0.124 | 0.092 | 0.125 | 0.165 | 0.21 | 0.223 |
| | Yes | Yes | 0.1 | 0.082 | 0.087 | 0.094 | 0.106 | 0.127 | 0.09 | 0.126 | 0.17 | 0.217 | 0.228 |
| RegNet | No | No | - | 0.083 | 0.095 | 0.108 | 0.129 | 0.155 | 0.11 | 0.141 | 0.174 | 0.209 | 0.22 |
| | Yes | No | - | 0.088 | 0.101 | 0.118 | 0.146 | 0.169 | 0.114 | 0.146 | 0.179 | 0.215 | 0.228 |
| | Yes | Yes | 0.1 | 0.083 | 0.092 | 0.102 | 0.12 | 0.145 | 0.097 | 0.122 | 0.16 | 0.195 | 0.206 |
| ViT-b-16 | No | No | - | 0.095 | 0.103 | 0.116 | 0.12 | 0.112 | 0.117 | 0.12 | 0.115 | 0.135 | 0.144 |
| | Yes | No | - | 0.096 | 0.107 | 0.117 | 0.111 | 0.096 | 0.122 | 0.123 | 0.118 | 0.138 | 0.144 |
| | Yes | Yes | 0.1 | 0.114 | 0.116 | 0.114 | 0.114 | 0.112 | 0.13 | 0.135 | 0.128 | 0.129 | 0.134 |

Table 11: Anomaly detection performance (AUROC) of ResNet, RegNet, and ViT-b-16 models trained using ImageNet.

| Architecture | Anchoring? | Zero-crop? | $\alpha$-value | LSUN (C) | LSUN (R) | iSUN | Textures | Places365 | NINCO |
|---|---|---|---|---|---|---|---|---|---|
| ResNet-18 | No | No | - | 97 | 95.07 | 95.35 | 86.25 | 80.9 | 75.76 |
| | Yes | No | - | 96.16 | 93.29 | 93.86 | 86.79 | 80.63 | 75.67 |
| | Yes | Yes | 0.025 | 96.88 | 92.27 | 92.49 | 86.42 | 79.94 | 76.4 |
| | Yes | Yes | 0.05 | 95.59 | 90.36 | 91.07 | 87.19 | 80.18 | 75.27 |
| | Yes | Yes | 0.1 | 94.94 | 92.37 | 93.02 | 86.14 | 79.93 | 75.03 |
| RegNet | No | No | - | 98.79 | 97.61 | 97.77 | 88.37 | 83.03 | 80.18 |
| | Yes | No | - | 98.77 | 98.04 | 98.01 | 87.6 | 83.39 | 80.44 |
| | Yes | Yes | 0.1 | 97.93 | 95.68 | 95.95 | 88.78 | 82.92 | 79.18 |
| ViT-b-16 | No | No | - | 91.59 | 87.34 | 86.92 | 79.24 | 65.72 | 65.98 |
| | Yes | No | - | 89.26 | 85.05 | 85.32 | 78.68 | 68.47 | 69.66 |
| | Yes | Yes | 0.1 | 89.42 | 86.37 | 85.81 | 78.99 | 67.87 | 70.79 |

refer to this method as *Hard Δ Mining*. Our idea conceptually aligns with the outlier exposure free OOD detection method used for object detection (Du et al., 2021). Figure 4 illustrates the trade off between the generalization and anomaly detection with decreasing regularization weight $\lambda$ for a fixed $\alpha = 0.25$ on the CIFAR-10 dataset. We find that with the introduction of the tails with higher regularization weights can improve anomaly detection (AUROC) at the cost of corruption accuracy.

While these strategies provide the motivation and capabilities to explore improved anchored training protocols, there are still open research questions for better refining our training protocol to achieve superior generalization while not compromising on safety metrics. A detailed study of which we leave for future work.

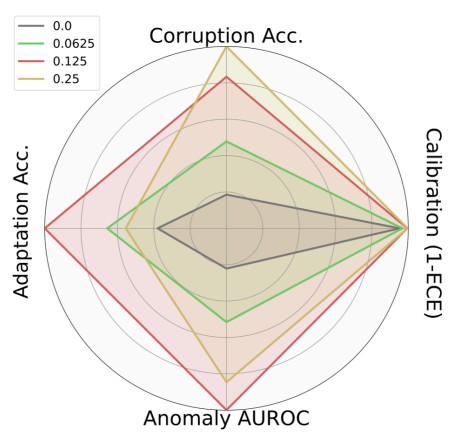

(a) Impact of masking probability $\alpha$ on generalization and safety metrics

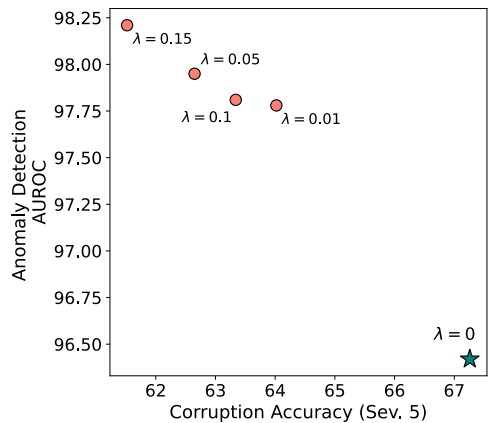

(b) Hard $\Delta$ Mining to control the Anomaly Detection and Corruption Accuracy for a fixed $\alpha = 0.25$

Figure 4: **Strategies to better understand the generalization vs safety trade-offs in anchored models**