# OpenReview forum: "Improving Generalization and Safety of Deep Neural Networks with Masked Anchoring"
_ICLR.cc/2024/Conference — Submitted to ICLR 2024_

### Official Review · Reviewer_8fzY · 2023-10-28

**Soundness:** 3 good
**Presentation:** 3 good
**Contribution:** 3 good
**Rating:** 6
**Confidence:** 3

**Summary:**

The authors explore the anchoring’s role in the generalization and safety properties of models. Based on this insight, the authors uncover that anchoring can be a training mechanism to improve generalization and safety.

**Strengths:**

The authors show the relationship between generalization ability and residual sensitivity.

This paper is well-written and easy to follow, and the method is simple and implementable.

The experimental results are extensive.

**Weaknesses:**

Masking probability $\alpha$ is a hyperparameter in the algorithm, which is difficult to estimate in practice.

Masking can reduce the sensitivity of the model to residual, but why can masking promote the smoothness of the model?

**Questions:**

Please refer to the questions in the Weaknesses.

---

> ### Author Response · Authors · 2023-11-18
> **Author Response - Reviewer 8fzY**
>
> Thank you for the positive assessment of our work. If our responses sufficiently address all the concerns raised, we request you to improve the score and champion our paper.
>
> > **Choice of hyper-parameter $\alpha$:**
>
> + The parameter $\alpha$ directly controls the schedule of residual corruption during training. Simply put, arbitrarily increasing $\alpha$ will start to adversely affect the training convergence and eventually the ID performance itself. In our study, we selected it solely based on two factors: We wanted to use a high value of $\alpha$ (for improved generalization), such that it can converge in the same number of epochs as that of anchoring w/o RAM, and can produce validation accuracies comparable (or better) than the non-anchored model.
>
> + Based on our experiments with CIFAR-10, CIFAR-100 and ImageNet across multiple architectures (RegNet, ResNet, WRN40-2, DEIT, ViTb and SWINv2) $\alpha = 0.2$ is a recommended setting. Similar to any existing regularization strategy (e.g., Mixup) adopted for improving generalization, RAM with high $\alpha$ might have the risk of producing models with reduced sensitivity towards anomalous data (i.e., compared to anchoring w/o RAM). However, we find that with $\alpha = 0.2$, the anomaly rejection trade-off is minimal and interestingly, it can even lead to higher rejection AUROC scores using networks with higher capacity (e.g., SWINv2-B, updated Table 2 in the paper).
>
> > **How does RAM help anchoring?**
>
> Please check our detailed answer to this central question in the summary response. For completion, we include a summary here.
>
> + To clarify this, we begin by considering generalization to an OOD test sample. In the tuple $(\mathrm{r}, \Delta)$ for the test sample $\mathrm{x}_t$, the anchor $\mathrm{r} \in P(R)$ and it is possible that $\Delta = \mathrm{x}_t - \mathrm{r}$ can be novel to P($\Delta$). Thus, by exposing the model to more diverse combinations of $(\mathrm{r}, \Delta)$ during training, generalization can be improved. A naive way to improve the diversity of P($\Delta$) (or equivalently P($X, \Delta$)) is to consider a wider anchor distribution (i.e., $P(R) \supset P(X) $). However, it is non-trivial to characterize the anchor distribution and arbitrarily improve its diversity; and increasing anchor diversity will lead to a combinatorially larger space (anchor, residual) pairs, which can in turn require larger number of epochs to effectively converge.
>
> + We adopt an alternative approach to increasing diversity of P($\Delta$) by making the residuals noisy, and propose a simple implementation in the form of RAM. Given the inherent challenge in defining a suitable residual noise distribution (and inferring its hyperparameters), we define the noise distribution to be same as the anchor distribution itself, i.e., P(R) = P(N) = P(X), and implement it efficiently using the RAM regularizer. Formally, for a given tuple $(\mathrm{r}, \mathrm{x} - \mathrm{r})$, anchor masking zeroes out the anchor while keeping the residual fixed, i.e., $(\mathrm{0}, \mathrm{x} - \mathrm{r})$. Generally, the tuple for making a prediction for $\mathrm{x}$ with a zero anchor (note: zero vector is a valid anchor in our anchor distribution) will be written as $(\mathrm{0}, \mathrm{x})$. However, in anchor masking this can be interpreted as making a prediction using the zero anchor, but with a noised residual $\mathrm{x} + \mathrm{\epsilon}$ where $\mathrm{\epsilon} = -\mathrm{r}$ and $\mathrm{\epsilon} \in P(N)$.

---

> > ### Author Response · Authors · 2023-11-22
> > **Request**
> >
> > Can you please check our response (and the updated paper), and let us know if you had any additional questions that we could answer before the discussion phase ends. Also, if you find our responses satisfactory, will you be open to raising the score to help champion our paper?

---

### Official Review · Reviewer_LRUp · 2023-11-02

**Soundness:** 4 excellent
**Presentation:** 3 good
**Contribution:** 4 excellent
**Rating:** 8
**Confidence:** 3

**Summary:**

The authors propose an extension to the method of anchoring, Random Anchor Masking (RAM), based on their empirical evaluation of anchoring on more typical Deep Neural Networks (DNNs) and tasks than it has been evaluated on in the past.

Anchoring is a recently proposed methodology designed to improve uncertainty estimation of DNNs by representing input samples as the difference (i.e. residuals) from a different randomly chosen sample within the training set (the anchor). As such, when used at inference time, anchored models are better able to estimate uncertainty since the residuals of an input not within distribuition are likely to be quite large. While anchoring has shown some success in uncertainty estimation on small scale benchmarks, the authors highlight that it has not been evaluated as much on more typical DNN architectures/problems, nor has it been evaluated in the context of other problems where distribution shifts occour, such as outliers, generalization under distribution shift and linear probing, calibration, robustness and outlier rejection. The authors proceed to evaluate standard anchoring on these problems with CIFAR-10/CIFAR-100 and ImageNet dataset variants relevant to the problem and on ResNet/WideResNet CNNs and Vision Transformers.

Despite the relatively good performance of anchoring on these problems, the authors are concerned that when the residual is large, such as would be expected in distribution shift, an anchored model may suffer significantly decreased generalization because it overfits the anchor. The authors propose to extend anchoring by simply masking (i.e. zeroing) the anchor (after residual calculation) with a probability $\alpha$. In other words, they dropout the anchor with probability $\alpha$, introducing noise in the anchored training process. For a small $\alpha$ this might be expected to reduce any dependence the model might learn on the anchors themselves and improve generalization on slightly OOD samples. The authors demonstrate results in a similar empirical analysis as used for standard anchoring and compare results with anchored and non-anchored models. Finally, the authors highlight that their results often show a trade-off for RAM/anchoring with generalization and each problem they evaluated, e.g. calibration. They describe how the hyper parameter $\alpha$ affects this tradeoff, and propose some methodology to mitigate it.

**Strengths:**

* Overall a very well-written and presented paper (with the notable exception of figure captions/labels).
* Clear motivation and clear method, although some of the design decisions for the method could be explained better.
* Very good background on explaining anchoring and why it might be interesting for other problems with distribution shift as an underlying cause/factor.
* Results are overall presented well in tables, and clear, again with the exception that some aspects are never explained (i.e. +/- is what exactly).
* The paper doesn't just propose a new anchoring method, but also evaluates the existing anchoring methodology on a wide range of problems within the deep learning literature, including robustness, calibration, OOD samples, corruption and linear probing/adaption.
* Reasonable evaluation of the proposed method, and existing anchoring methodology, using appropriate datasets (CIFAR-10/CIFAR-100/ImageNet, and the task variants of these) and a wide range of DNN models, including ResNet/WideResNet CNNs and a Vision Transformer.
* Almost all of the presented results for RAM beat the baselines models modestly with and without anchoring, with perhaps the exception of calibration which is much closer. Overall the results across a wide range of problems are quite impressive, again compared to the baselines.
* The analysis on the trade-off for generalization v.s. each task, e.g. calibration, is welcome and in particular Figure 2 is great to give a high-level overview of what this tradeoff is for some variants of the hyper parameter $\alpha$.

**Weaknesses:**

* Why this type of added noise during training v.s. say residual noise, or other noise applied to the mask? While the intuition of noisy training reducing the model overfitting on anchors makes sense, the intuition as to why the authors believe anchor overfitting is a problem isn't completely obvious to me.
* RAM is a very unfortunate acronym to be using in computer circles, while I appreciate the acronym isn't made up and is meaningful, it would probably be best to come up with another one.
* Not clear to me if the same $\alpha$ is applied to the whole batch, or is per-sample. The text in section 3.1 makes it sounds like the former, but that doesn't make sense to me, it seems very strange to mask out a whole batch's anchors instead of a random subset, e.g. like dropout.
* Calibration results appear to be relatively weak, with many of the results presented being within the variance(?) of baselines.
* The captions of figures/tables and labelling of figures is weak, with little explanation to be found within a figure's caption, and labels missing from elements of figures - e.g. missing $\alpha$ on the legend of Figure 2(a). Ideally figures and tables should almost be self-contained, but I found myself having to read the text quite a bit to figure out each one.
* While the authors compare the performance of RAM w.r.t. the models with and without anchoring on standard task-specific datasets (here I use task to loosely mean problems such as calibration, corruption, robustness, etc). For example while using corrupted ImageNet-C, they do not compare with state-of-the-art methods on that dataset. I personally am willing to forgive this, as the authors do not claim state-of-the-art results on these tasks, and appear to be the first to evaluate anchoring on these tasks at all.
* Little discussion apparently on the effect of adding noise to training on the training time of the model, typically when we add noise to training and make it more difficult, it might be expected that training would take longer to reach the same generalization.
* Not clear how the $\alpha$ hyperparameter values found throughout the paper are decided on

**Questions:**

* Is $\alpha$ is applied to the whole batch, or is it per-sample? In 3.1 "...RAM completely masks the anchors in the tuple for a given batch during training...". If it is for the whole batch, why would you do it this way instead of per-sample, which aligns more with the objective of adding some noise to training (i.e. dropout).
* What is the +/- shown in results, as I don't see it explained anywhere in the paper, i.e. is it variance, stddev, ...? Ideally this should be explained in the captions of the figures themselves.
* Is RAM close to state-of-the-art on any of the datasets it is evaluated with, e.g. ImageNet-C, etc? If not, how far away from SOTA is it, and what do you think it would take to bring anchoring closer to SOTA in these tasks?
* You claim no computational overhead in section 4.2 for RAM w.r.t. anchoring, which I believe given the method. However, anchoring itself does have a computational overhead over standard methods of training since the first (and most computationally expensive) layer of e.g. a CNN is effectively doubled in size. What is the trade-off on this front? Is the tradeoff reasonable for large models still, i.e. ViT?
* Furthermore, since we are making training harder vis-a-vis dropout, what is the increase in training time to reach the same generalization as anchored models? What about compared to models without anchoring?

---

> ### Author Response · Authors · 2023-11-18
> **Author Response - Reviewer LRUp**
>
> Thank you for your positive feedback. Here is our detailed response to all your questions.
>
> **1. Choice of Acronym RAM:**
>
> We understand how this terminology can be confusing. Following reviewer advice, we will use MAR (Masking Anchors Randomly) as the new acronym. We have not changed it yet so that the other reviewers do not get confused.
>
> **2. Implementation of RAM:**
>
> The reviewer is right in pointing out that the parameter $\alpha$ controls the schedule of residual corruption during training -- across an entire batch. However, this is only an implementation choice and one can consider alternative approaches, and the observations will not change. For example, the residual corruption can be applied to $\alpha \%$ of samples from each batch or this can be included as an additional loss objective with a pre-specified penalty.
>
> **3. Complexity of Anchored Training w/o and w/ RAM:**
>
> From an optimization standpoint, anchored training becomes increasingly challenging as the combinatorial space of (anchor, residual) pair grows. For example, with CIFAR-10 or CIFAR-100, anchored training converges effectively with the same training recipe in the same number of epochs as the standard training. However, with ImageNet (regardless of the architecture), for the same training recipe, anchored training requires $20$ additional epochs to converge to the same level of validation loss. However, the training efficiency itself is not significantly different from non-anchored models. Note that, including RAM regularization ($\alpha = 0.2$ in all experiments) does not impact the training behavior and requires the same number of epochs as standard anchoring. Here we show the comparisons for the SWINv2-B model trained on ImageNet.
>
>
> |     Model     | Anchoring? | RAM? | # Epochs | Model Size (MB) | Time (s) ImageNet-Test  (16 GPUs) | ID Acc. | OOD Acc.ImageNet-C (Sev 5.) |
> |:-------------:|:----------:|:----:|:--------:|:---------------:|:---------------------------------:|:-------:|:---------------------------:|
> |               |     No     |  No  |    300   |      336.37     |                33.0               |  84.06  |             45.7            |
> | SWINv2 (base) |     Yes    |  No  |    320   |      336.40     |                38.0               |  84.11  |             47.6            |
> |               |     Yes    |  Yes |    320   |      336.40     |                38.0               |  84.09  |            **48.16**            |
>
>
> **4. Improving captions of Tables and Figures:**
>
> Thanks for this comment. We have expanded the figure and table captions for better clarity.
>
> **5. Hyperparameter Selection:**
>
> For all experiments reported in the paper, we follow the training recipes from the torchvision (https://pytorch.org/vision/stable/models.html) and directly used the same hyper-parameter configurations. The only difference was increasing the number of epochs for the case of imagenet ($20 more epochs).
>
> For the hyperparameter $\alpha$, we selected it solely based on two factors: We wanted to use a high value of $\alpha$ (for improved generalization), such that it can converge in the same number of epochs as that of anchoring w/o RAM, and can produce validation accuracies comparable (or better) than the non-anchored model. Based on our experiments with CIFAR-10, CIFAR-100 and ImageNet across multiple architectures (RegNet, ResNet, WRN40-2, DEIT, ViTb and SWINv2) $\alpha = 0.2$ is a recommended setting.
>
> **6. What is the $\pm$ shown in results?**
>
> We apologize for this oversight. $\pm$  in the calibration metric indicates the standard deviation across all OOD datasets (ImageNet-C, Imagenet-$\bar{\text{C}}$, ImageNet-v2, ImageNet-R and ImageNet-S) and we have made this clear in the updated version.
>
>
> **7. SoTA Comparison:**
>
> The key claim of this work is that, across dataset sizes (CIFAR-10 to Imagenet), architectures (ResNet to ViT) and network sizes (5M to 88M parameters), anchored training can provide significant gains in OOD generalization, anomaly rejection and adaptation, compared to vanilla training. We do not attempt to establish SoTA generalization performance, since top results are often obtained via model souping[1] or by fine-tuning large scale pre-trained models[2]. However, we emphasize that, when the train recipe includes high-capacity architectures or advanced training strategies (Mixup, EMA, label-smoothing, cutmix), anchoring continues to improve over the base models in all these cases, showing promise to be effective even with SoTA recipes involving large-scale pre-training and ensembling. In the updated conclusions, we have highlighted that integrating anchoring (w/ RAM) into these SoTA approaches is an important future direction of work.
>
> [1] M. Wortsman et al. Model soups: averaging weights of multiple fine-tuned models improves accuracy without increasing inference time, ICML 2022
>
> [2] S.Goyal et al. Finetune like you pretrain: Improved finetuning of zero-shot vision models, CVPR 2023.

---

> > ### Comment · Reviewer_LRUp · 2023-11-22
> >
> > I'd like to thank the authors for their rebuttal, and apologize for my late participation in the rebuttal period - this was due to exceptional circumstances.
> >
> > I believe the authors have addressed most of my questions/concerns, and hopefully by including the better figures and descriptions of some aspects of their methodology, this has also strengthened the paper significantly.

---

> > > ### Author Response · Authors · 2023-11-22
> > > **Thank you!**
> > >
> > > We appreciate you taking the time to check our response!

---

### Official Review · Reviewer_GTiz · 2023-11-05

**Soundness:** 3 good
**Presentation:** 3 good
**Contribution:** 3 good
**Rating:** 6
**Confidence:** 4

**Summary:**

This paper investigates a recent architecture named anchoring.  By evaluating important tasks of out-of-distribution generalization, task adaptation, anomaly detection, and calibration, the authors give a comprehensive study of anchoring for the safety and generalization of deep neural networks. Based on this, a simple modification called Random Anchor Masking (RAM) is proposed and it shows significant performance gains over both standard anchoring and non-anchored models. Extensive experiments across datasets of varying size and complexity (CIFAR10- 100/ImageNet) as well as architectures of varying scales (RegNet/ResNet/WRN/ViT) are conducted, which verify the effectiveness of the proposed RAM methods.

**Strengths:**

1. This paper investigates a new and novel topic named anchoring, from the perspective of improving the generalization and safety of deep neural networks.
2. This paper is well-written and easy to follow.
3. The proposed method is simple but effective, and it gives some insight by linking the improved generalization behavior of anchored models to their sensitivity to the space of residuals.
4. Extensive experiments across datasets of varying size and complexity (CIFAR10- 100/ImageNet) as well as architectures of varying scales (RegNet/ResNet/WRN/ViT) are conducted, which verify the effectiveness of the proposed RAM methods.

**Weaknesses:**

1. The proposed RAM method aims at improving the smoothness of the relationship between residual and target labels. However, why does such an easy strategy work on so many datasets and applications? It seems that more analysis and explanations should be given, apart from the extensive experiments showing that it works.
2. According to the impact of masking probability alpha on generalization and safety metrics, it shows a direct correlation between alpha and corruption accuracy. However, why such a correlation is held? Will this correlation be held for more datasets or applications?

**Questions:**

Please see the weaknesses section.

**Details Of Ethics Concerns:**

No ethics review is needed.

---

> ### Author Response · Authors · 2023-11-18
> **Author Response - Reviewer GTiz**
>
> Thank you for a positive assessment of our paper. If our response sufficiently addresses your questions, we request you to improve the score and champion our paper.
>
> **1. Why does anchored training and the RAM regularizer lead to better generalization?**
>
> Please check our detailed answer to this central question in the overall response. For completion, we include a summary here. While we have not theoretically characterized the behavior of RAM, our hypothesis is rooted in existing theory and our expanded empirical results provide clear evidence for the hypothesis. However, motivated by the empirical success of anchoring, we plan to carry out a theoretical study on generalization in anchored models as part of our future work (as specified in the updated Conclusion section).
>
> -   In essence, anchoring involves representing an input $\mathrm{x}$ as a combination of an anchor $\mathrm{r}$ and the residual $(\mathrm{x} - \mathrm{r})$, while maintaining all other aspects of the conventional neural network. This method capitalizes on the lack of shift invariance in Neural Tangent Kernels (NTKs) induced by common neural networks. Anchored training employs different anchors for the same sample across epochs to marginalize the effect of anchor choice during inference. While it may seem akin to data augmentation, anchoring is fundamentally different in that it combinatorially expands the space of (anchor, residual) pairs with different anchor choices and enables exploration of a richer class of hypotheses, and does not manipulate the inputs. However, as our experiments show, anchoring can be combined with any additional data augmentation technique.
>
> -   Now, let us consider generalizing to an OOD test sample. In the tuple $(\mathrm{r}, \Delta)$ for the test sample $\mathrm{x}_t$, the anchor $\mathrm{r} \in P(X)$ and it is possible that $\Delta = \mathrm{x}_t - \mathrm{r}$ can be novel to $P(\Delta)$. Thus, by exposing the model to more diverse combinations of $(\mathrm{r}, \Delta)$ during training, generalization can be improved. A naive way to improve the diversity of $P(\Delta)$ (or equivalently $P(X, \Delta)$) is to consider a wider anchor distribution (i.e., $P(C) \supset P(X) $). However, it is non-trivial to characterize the anchor distribution and arbitrarily improve its diversity; and increasing anchor diversity will lead to a combinatorially larger space (anchor, residual) pairs, which can in turn require larger number of epochs to effectively converge.
>
> -   We adopt a simpler alternative by making the residuals noisy, and propose a simple implementation in the form of RAM. Given the inherent challenge in defining a suitable noise distribution (and inferring its hyperparameters), we define the noise distribution to be same as the anchor distribution itself, i.e., $P(C) = P(N) = P(X)$, and implement it efficiently using the RAM regularizer. Formally, for a given tuple $(\mathrm{r}, \mathrm{x}-\mathrm{r})$, anchor masking can be reinterpreted as $(\mathbf{0}, \mathrm{x} + \epsilon)$ where $\epsilon \in P(N)$. In other words, we want to identify the true label for $\mathrm{x}$ with the zero anchor, given the noisy residual.
>
> -   Finally, network capacity plays a significant role in effectively leveraging the increased diversity produced by RAM. For example, with ImageNet, as we move from RegNet (5M) to SWINv2-B (88M), we witness larger performance improvements over both anchoring w/o RAM as well as standard training (see updated Table 1 in the paper).
>
> **2. Choice of $\alpha$ and the Generalization-Anomaly Rejection Trade-off:**
>
> -   The parameter $\alpha$ directly controls the schedule of residual corruption during training. Simply put, arbitrarily increasing $\alpha$ will start to adversely affect the training convergence and eventually the ID performance itself. Hence, we recommend a nominal value of $\alpha=0.2$, which we empirically find to converge in the same number of epochs as anchoring w/o RAM across all architectures, and to result in much improved generalization.
>
> -   Similar to any existing regularization strategy (e.g., Mixup) adopted for improving generalization, RAM with high $\alpha$ has the risk of producing models with reduced sensitivity towards anomalous data (i.e., compared to anchoring w/o RAM) [1]. We find that with $\alpha = 0.2$, the anomaly rejection trade-off is minimal and interestingly, it can even lead to higher rejection AUROC scores with networks with higher capacity (e.g., SWINv2-B, updated Table 2 in the paper).
>
> [1] Hendrycks, Dan, et al. "Pixmix: Dreamlike pictures comprehensively improve safety measures.", CVPR 2022.

---

> > ### Author Response · Authors · 2023-11-22
> > **Request**
> >
> > Can you please check our response (and the updated paper), and let us know if you had any additional questions that we could answer before the discussion phase ends. Also, if you find our responses satisfactory, will you be open to raising the score to help champion our paper?

---

> > > ### Comment · Reviewer_GTiz · 2023-12-03
> > > **Response To Authors**
> > >
> > > Thanks for providing such a detailed response. After reading reviews from other reviewers and the author's response,  I decided to maintain my score.

---

### Official Review · Reviewer_auhR · 2023-11-09

**Soundness:** 2 fair
**Presentation:** 2 fair
**Contribution:** 2 fair
**Rating:** 5
**Confidence:** 4

**Summary:**

In this paper, the authors try to provide a high-level understanding of the advantages of the anchoring method, which is believed to offer the enhanced generalization capability of deep models. Also, the authors propose a variant of the popular anchoring method, which masks the anchor sample to achieve further improvements in generalization. Their simple yet effective method is widely evaluated on various tasks and model architectures to confirm the advantages of their strategy.

**Strengths:**

**Strength 1**: The authors try to provide a comprehensive benefit of anchoring by evaluating the set of related tasks, which have been done separately but are commonly related to the generalization capability of models. Specifically, the corruption robustness benchmarks, including a set of `-C' tasks, the anomaly detection benchmarks, and the various kinds of downstream transfer learning tasks, are largely considered. I believe that the extensive and collective evaluations can provide a comprehensive understanding of the strength of the anchoring methods.

**Weaknesses:**

**Weakness 1:** The main motivation of this work seems to offer a 'clear' understanding of the benefits of anchoring, but I feel that the provided explanation is not sufficient to understand anchoring thoroughly. The main reason for this critique is that the analysis remains empirical observations rather than trying to provide theoretical analysis. Most of the researchers conjecture that random perturbations on inputs, e.g., data augmentations with noise or corruptions, or the interpolations between samples, e.g., a family of methods with Mixup such as Mixup, Manifold mixup, CutMix, etc, are linked to the improved generalization of models by letting models face the diversity beyond clean data sample-wise training. However, pushing the 'high-level conjecture' forward 'theoretical proofs' is crucial. I feel that this work has tried to do that, but is quite limited in providing the empirical observations and conjectures, without theoretical proofs.

**Weakness 2:** It is hard to find an explicit logical link between the proposed strategy (i.e., Random Anchoring Masking) and the understanding of anchoring. Why do we have to randomly mask the anchor in order to improve the generalization? Are there any further insights into the proposed strategy beyond the simple objective of imposing higher diversity on models?

**Weakness 3:** The paper is not well organized. For example, it would have been better to organize Related Work to clearly analyze the previous works on both technical and theoretical sides. I know that the author's analyses of the prior works are spread into multiple sections in this version of the article. However, it is better to show a designated section to emphasize the uniqueness of this work beyond prior efforts is strongly required. Also, I guess that the Conclusion section is missing.

There are some minor corrections, including typos.
- In '2.1 Preliminaries', at the 'Notations' part: "of distinct classes.." -> "of distinct classes."
- In the third sentence of '3. An Anchoring Perspective to Generalization': "it was was" -> "it was"

**Questions:**

**Q1:** As pointed out in 'Weakness 1', would you provide any theoretical support to clearly understand the advantages of anchoring and the proposed masked anchoring method?

**Q2:** As aforementioned in 'Weakness 2', would you provide the logical reasoning of the masking-based approach for anchoring?

---

> ### Author Response · Authors · 2023-11-18
> **Author Response - Reviewer auhR**
>
> Thank you for your comments and questions. Following your feedback, we have expanded the background, reorganized the results and clearly stated our findings. If our responses sufficiently address all the concerns raised, we request you to improve the score and champion our paper.
>
> **1. How does anchoring work and why does RAM lead to better generalization?**
>
> In order to clarify how anchoring works and why the proposed RAM regularization leads to improved generalization, we have provided a detailed explanation in the overall response. Here we include an expanded version. While we have not theoretically characterized the behavior of RAM, our hypothesis is rooted in existing theory and our expanded empirical results provide clear evidence for the hypothesis. However, motivated by the empirical success of anchoring, we plan to carry out a theoretical study on generalization in anchored models as part of our future work (see updated Conclusion section).
>
> + In regard to theoretical foundations for anchoring, we directly refer to two key results: (a) In [1], it was shown that **centering a dataset using different constant inputs will lead to different solutions**, due to inherent lack of shift invariance in NTKs induced by commonly adopted neural networks. Anchored training uses different anchors for the same sample across different epochs with the goal of marginalizing out the effect of anchor choice at inference time. But in this process, it implicitly explores a large class of hypotheses and enforces prediction consistency; (b) In [2], it was theoretically showed that an anchored model can **demonstrate improved extrapolation behavior at test time, when the anchor $\mathrm{x} \in P(X)$ and the residual for the unseen sample $\Delta \in P(\Delta)$**. However, neither of these works rigorously studied the generalization behavior of anchored models on large-scale datasets, architectures or even practical distribution shifts. We have also accordingly updated the background section.
>
> + While the idea of enforcing prediction consistency across different anchor choices might appear similar to data augmentation, we want to clarify that **anchoring does not impose any invariance to data characteristics**, but only expands the combination of (anchor, residual) pairs with each additional anchor.
>
> + For the first time, we systematically study the viability of anchoring as a training protocol across dataset sizes (CIFAR-10 to Imagenet), architectures (ResNet to ViT) and network sizes (5M to 88M parameters). In this context, we also study other key properties such as calibration, anomaly rejection and adaptation (both ID and OOD evaluation). We have entirely reorganized the results section for clarity.
>
> + Let us consider generalizing to an OOD test sample. In the tuple $(\mathrm{r}, \Delta)$ for the test sample $\mathrm{x}_t$, the anchor $\mathrm{r} \in P(X)$ and it is possible that $\Delta = \mathrm{x}_t - \mathrm{r}$ can be novel to $P(\Delta)$. Thus, by **exposing the model to more diverse combinations of $(\mathrm{r}, \Delta)$ during training, generalization can be improved**. A naive way to improve the diversity of $P(\Delta)$ (or equivalently $P(X, \Delta)$) is to consider a wider anchor distribution (i.e., $P(C) \supset P(X) $). However, it is non-trivial to characterize the anchor distribution (e.g., for ImageNet) and arbitrarily improve its diversity; and increasing anchor diversity will lead to a combinatorially larger space of (anchor, residual) pairs, thus requiring a larger number of epochs to effectively converge.
>
> + We adopt an alternative approach to increasing diversity of $P(\Delta)$ by **making the residuals noisy, and propose a simple implementation in the form of RAM**. Given the challenge in defining a suitable residual noise distribution, we define the noise distribution to be same as the anchor distribution itself, i.e., $P(C) = P(N) = P(X)$. Formally, for a given tuple $(\mathrm{r}, \mathrm{x}-\mathrm{r})$, anchor masking zeroes out the anchor while keeping the residual fixed, i.e., $(\mathbf{0}, \mathrm{x}-\mathrm{r})$. Generally, the tuple for making a prediction for $\mathrm{x}$ with a zero anchor (note: zero vector is a valid anchor) will be written as $(\mathbf{0}, \mathrm{x})$. However, anchor masking can be interpreted as **making a prediction using the zero anchor, but with a noised residual $\mathrm{x} + \epsilon$** where $\epsilon \in P(N)$.
>
> + Finally, network capacity plays an important role in effectively leveraging the increased diversity produced by RAM. For example, with ImageNet, as we move from RegNet (5M) to SWINv2-B (88M), we witness larger generalization performance improvements over both anchoring w/o RAM as well as standard training (see updated Table 1).
>
> [1] J.J. Thiagarajan et al. Single model uncertainty estimation via stochastic data centering, Neurips, 2022.
>
> [2] A. Netanyahu et al. "Learning to Extrapolate: A Transductive Approach." ICLR 2023.

---

> > ### Author Response · Authors · 2023-11-22
> > **Request**
> >
> > Can you please check our response (and the updated paper), and let us know if you had any additional questions that we could answer before the discussion phase ends. Also, if you find our responses satisfactory, will you be open to raising the score to help champion our paper?

---

> > > ### Comment · Reviewer_auhR · 2023-11-23
> > > **Thanks for the response**
> > >
> > > Dear Authors,
> > >
> > > I truly appreciate the kind response and careful revision of the manuscript.
> > > First of all, I feel that the revised manuscript is much easier to follow and shows a clear logical flow to reach the proposed algorithm. Again, thanks to the revision.
> > > Also, your insights on the reasons for gains from RAM allow me to capture a high-level understanding of why RAM improves performance. However, as you responded, it seems to remain at a conjecture level but does not reach a theoretical level.
> > > I can say that the additional explanation indeed promotes a better understanding of readers, but I am keeping to expect this work to provide further in-depth theoretical analysis. For example, as you described, [2] provides a theoretical foundation for the improved extrapolation behavior of anchoring; then, what happens for RAM? I guess that RAM is a simple manipulation of the anchor pair, so it is probably straightforward to further formulate the extrapolation behavior of RAM by stepping on the results from [2]. Albeit the weakness, I slightly leaned to increase my rating due to the improvements from your revision, but I have a raising concern as follows.
> > >
> > > In the revised manuscript, a set of more complicated ViT-based architectures is largely considered. Unfortunately, when seeing the results in Table 1-4, the performance gains of RAM seem to be ambiguous. It contrasts with clear gains in ResNet architectures (Table 5).
> > > For OOD (Table 1), RAM generally seems to be the best. But for the Calibration test (Table 2), RAM achieves the top for 19 cases, naive anchoring achieves 15 cases, and non-anchoring takes the remaining 8 cases; which makes me worry about the dominance of RAM over baselines. For the LP-based adaptation on ImageNet-1K, RAM achieves the best 'Average' for 3 backbone cases, but seems to be inferior to others for the remaining two backbone cases; it also seems to be ambiguous gains. Lastly, the LP-based adaptation on DomainNet, RAM does not seem to be the best; it achieves a few numbers of top accuracies in 'Average'.
> > > The new concern makes me worry about the applicability of RAM to larger architectures, and curious about the reasons for different behavior between the ResNet group and the ViT group.
> > >
> > > By considering all the positive and negative viewpoints, I want to increase my rating to 5 (marginal below the acceptance) from 3 (reject). Again, I really appreciate your extensive efforts to revise your manuscript.

---

### Official Review · Reviewer_rVkq · 2023-11-09

**Soundness:** 3 good
**Presentation:** 3 good
**Contribution:** 3 good
**Rating:** 6
**Confidence:** 3

**Summary:**

This paper extends the anchoring-based training of deep neural networks with a Random Anchor Masking (RAM) technique. The authors observed that this simple RAM strategy can significantly improve multiple metrics under domain shift setting. The empirical evaluation is conducted on CIFAR-10/100 and ImageNet datasets, which shows the positive results by considering the RAM technique in anchoring-based training. Another observation is the trade-off between OOD generalization and anomaly detection, demonstrating that models that achieved superior gains in generalization performance tend to exhibit a trade-off in their ability to detect OOD samples. The authors investigated the effect of masking ratio $\alpha$ on this trade-off.

**Strengths:**

1. The study of anchoring-based training and inference is interesting. Especially, there are some evidences that this training strategy can benefit some safety-aware metrics.

2. The proposed masking-based strategy is simple to implement. The empirical evaluation shows the benefits of this simple strategy.

3. The illustrative experiment in Figure 1 shows clear motivation for the proposed strategy, where models training by anchoring+RAM seem can better estimate the uncertainty for corrupted samples.

**Weaknesses:**

1. RAM proposed in this paper is very simple and easy to implement, which is a great advantage. However, the principle and explanation behind this simple strategy are essential. The authors have provided some motivation for this masking strategy, but further explanation of the underlying principles can further improve the quality of the article. For example, the authors mentioned that "this noisy training induced by RAM enforces the model to improve smoothness between the target and the residuals leading to better calibrated predictions". How RAM improves smoothness and how this smoothness leads to better calibration can be further explained.

2. For some other findings in the experiment, such as the trade-off between generalization and OOD detection, this paper only provided some simple experimental results. The reasons behind these phenomena were not deeply analyzed. Is this phenomenon also present in other methods?

3. In inference, multiple random anchors can be used to obtain predictions and the variance can be interpreted as an estimate of the epistemic uncertainty. This is very similar with the simple deep ensemble strategy, the comparison with ensemble-based methods could be provided.

Minor issues: "was was argued" ==> "was argued", "results in 2" ==> "results in Table 2", "Figure 3a depicts" ==> "Figure 2a depicts".

**Questions:**

Please refer to weaknesses section.

---

> ### Author Response · Authors · 2023-11-18
> **Author Response - Reviewer rVkq**
>
> Thank you for your positive comments and the questions. If our responses sufficiently address all the concerns raised, we request you to improve the score and champion our paper.
>
> **How does anchoring work and why does RAM lead to better generalization?**
>
> In order to clarify how anchoring works and why the proposed RAM regularization leads to improved generalization, we have provided a detailed explanation in the overall response. Here we include an expanded version of that with some additional comments to address your questions. While we do not theoretically establish the utility of RAM, our hypothesis is rooted in existing theory and our expanded empirical results provide clear evidence for the hypothesis.
>
> -   In essence, anchoring involves representing an input $\mathrm{x}$ as a combination of an anchor $\mathrm{r}$ and the residual $(\mathrm{x} - \mathrm{r})$, while maintaining all other aspects of the conventional neural network. This method capitalizes on the **lack of shift invariance in Neural Tangent Kernels (NTKs)** induced by common neural networks. Anchored training employs different anchors for the same sample across epochs to marginalize the effect of anchor choice during inference.
>
> + While **it may seem akin to data augmentation, anchoring is fundamentally different** in that it combinatorially expands the space of (anchor, residual) pairs with different anchor choices and does not manipulate the inputs. However, as our experiments show, anchoring can be combined with any additional data augmentation technique.
>
> -   For the first time, we systematically study the viability of anchoring as a training protocol across dataset sizes (CIFAR-10 to Imagenet), architectures (ResNet to ViT) and network sizes (5M to 88M parameters). In this context, we also study other key properties like calibration, anomaly rejection and adaptation (both ID and OOD evaluation). We have entirely reorganized the results section for clarity.
>
> -   Now, let us consider generalizing to an OOD test sample. In the tuple $(\mathrm{r}, \Delta)$ for the test sample $\mathrm{x}_t$, the anchor $\mathrm{r} \in P(X)$ and it is possible that $\Delta = \mathrm{x}_t - \mathrm{r}$ can be novel to $P(\Delta)$. Thus, by exposing the model to more diverse combinations of $(\mathrm{r}, \Delta)$ during training, generalization can be improved. A naive way to improve the diversity of $P(\Delta)$ (or equivalently $P(X, \Delta)$) is to consider a wider anchor distribution (i.e., $P(C) \supset P(X) $). However, it is non-trivial to characterize the anchor distribution and arbitrarily improve its diversity; and increasing anchor diversity will lead to a combinatorially larger space (anchor, residual) pairs, which can in turn require larger number of epochs to effectively converge.
>
> -   We adopt an alternative approach to increasing diversity of $P(\Delta)$ by making the residuals noisy, and propose a simple implementation in the form of RAM. Given the inherent challenge in defining a suitable residual noise distribution (and inferring its hyperparameters), we define the noise distribution to be same as the anchor distribution itself, i.e., $P(C) = P(N) = P(X)$. Formally, for a given tuple $(\mathrm{r}, \mathrm{x}-\mathrm{r})$, anchor masking zeroes out the anchor while keeping the residual fixed, i.e., $(0, \mathrm{x}-\mathrm{r})$. Generally, the tuple for making a prediction for $\mathrm{x}$ with a zero anchor (note: zero vector is a valid anchor) will be written as $(\mathbf{0}, \mathrm{x})$. However, in anchor masking this can be interpreted as making a prediction using the zero anchor, but with a noised residual $\mathrm{x} + \epsilon$ where $\epsilon \in P(N)$.
>
> **2. Trade-off between OOD Generalization and Anomaly Rejection:**
>
> The parameter $\alpha$ controls the schedule of residual corruption during training. Arbitrarily increasing $\alpha$ can adversely affect the training convergence and eventually the ID performance itself. Hence, we recommend a nominal value of $\alpha=0.2$, which we empirically find to converge in the same number of epochs as anchoring w/o RAM across all architectures, and to result in much improved generalization. Similar to any existing approach (e.g., Mixup) adopted for improving generalization, RAM with high $\alpha$ could lead to reduced sensitivity towards anomalous data [1]. However, we find that with $\alpha = 0.2$, the anomaly rejection trade-off is minimal and interestingly, it can even lead to higher rejection AUROC scores with networks with higher capacity (e.g., SWINv2-B, updated Table 2).
>
> [1] Hendrycks, Dan, et al. "Pixmix: Dreamlike pictures comprehensively improve safety measures.", CVPR 2022.

---

> > ### Author Response · Authors · 2023-11-18
> > **Author Response (contd.)**
> >
> > **3. Comparison to Ensembling:**
> >
> > Thank you for this important question. While anchoring jointly trains with multiple anchors for the same sample and implicitly explores a richer class of hypotheses, it does not actually produce an ensemble. More importantly, the underlying principle of anchored training is to marginalize the effect of anchor choice at inference time. We perform multiple passes (i.e., we estimate the prediction as an average of 5 passes with different random anchors) for obtaining robust predictions (to account for any failure case, where the model failed to produce consistent predictions across all anchors).
> >
> > However, we want to emphasize that one can always build an ensemble of anchored models akin to conventional ensembles, and more importantly, we hypothesize that it can lead to a much richer set of features. To demonstrate this, we performed an additional experiment following this recent work [2], which showed that using the collection of features from an ensemble for linear probing leads to much superior adaptation performance under covariate and task shifts. Here, we construct an ensemble of ResNet-18 models for CIFAR-100 data and we make a surprising finding that anchored ensembles require 50\% fewer parameters than the standard ensemble for achieving the same adaptation accuracy. And with the same number of parameters, provide significant improvements.
> >
> > | **Model** | **Ensemble Size** | **UCF 101** | **Oxford Pets** |
> > |:---------:|:-----------------:|:-----------:|:---------------:|
> > |  Vanilla  |         3         |     43.7    |       33.7      |
> > |  Vanilla  |         6         |     46.0    |       35.5      |
> > |    Ours   |         3         |     49.1    |       37.5      |
> > |    Ours   |         6         |   **50.2**  |     **39.2**    |
> >
> > [2] Zhang, Jianyu, and Léon Bottou. "Learning useful representations for shifting tasks and distributions.", ICML 2023

---

> > > ### Comment · Reviewer_rVkq · 2023-11-21
> > > **Thanks for the response**
> > >
> > > Thanks for your response.
> > >
> > > - The first question is actually asking the intuition behind the proposed RAM rather than the basic anchoring training strategy. Despite reviewing the response, I still find it challenging to comprehend the intuition regarding how RAM enhances smoothness and how this improved smoothness contributes to better calibration.
> > >
> > > - The second question concerns the intuition what causes this observed trade-off. While the authors mention similarities to other approaches like Mixup, a direct answer to this question is not provided.
> > >
> > > - The results presented in the response for the third question appear to be impressive.
> > >
> > > Thank you again for taking the time to answer my questions. Based on the response, I would like to keep my score unchanged.

---

> > > > ### Author Response · Authors · 2023-11-21
> > > > **Additional Comments**
> > > >
> > > > We appreciate you taking the time to check our response. Hope our new response clarifies some of your persisting concerns.
> > > >
> > > > > **How does RAM help improve generalization?**
> > > >
> > > > We explained the anchoring strategy just to motivate RAM. Simply put, an OOD test sample (e.g., corruption) manifests as variations in the residual (anchor will be always from $P(R)$) and **by systematically considering noisy residual during training, RAM promotes robustness (or smoothness) to changes in $\Delta$**. In that regard, RAM **exposes the model to more variations of (anchor, residual) pairs**, while also making the training process converge well without requiring additional epochs. This is a practical alternative to arbitrarily increasing the anchor set, which can make the training more challenging.
> > > >
> > > > > **Trade-off between different  axes of safety**
> > > >
> > > > Given that the safety objectives are becoming more elaborate in practical AI systems (OOD generalization, adversarial robustness, anomaly rejection, calibration, prediction consistency etc.), it has become **imperative to understand if a given training strategy sacrifices along one of the safety axes to improve another**. While a rigorous theoretical framework is yet to be developed to characterize this,  several empirical studies have showed that the trade-off can be non-trivial in practice. For example, adversarial training improves adversarial robustness but sharply degrades other classifier performance metrics. Similarly, outlier exposure methods increase the anomaly rejection capabilities but suffer  when it comes to corruption accuracy. Finally, strong data augmentation and regularization techniques often improve OOD robustness but harm anomaly
> > > > detection, due to their tendency to over-generalize. In this regard, we systematically study the empirical behavior of anchoring w/ and w/o RAM across different axes and find that RAM leads to improved OOD generalization, calibration and consistency, with only **minimal comprise in anomaly rejection performance (i.e., does not over-generalize)**. Performing a more principled theoretical analysis of this behavior is definitely part of our future work.
> > > >
> > > > And finally, thanks for the feedback about the additional experiment.

---

### Official Review · Reviewer_BVnY · 2023-11-13

**Soundness:** 3 good
**Presentation:** 3 good
**Contribution:** 3 good
**Rating:** 6
**Confidence:** 3

**Summary:**

Authors present an overview on the benefits of anchoring for training deep models, and provide a novel technique named Random Anchor Masking (RAM) in which the anchor is masked with a given probability $\alpha$, thus only the residual $x-r$ is used for prediction. The empirical results show that with this approach better results are achieved than with standard anchoring.

**Strengths:**

- The proposed method does not require any changes to optimization, as it operates only in the input layer

- The experimental evaluation is performed on multiple datasets and architecture complexity

- The paper is well written

**Weaknesses:**

I am not an expert on anchoring, so the main issues I encouter in this work are:

- It is not trivial for me to understand why modelling $P(X,\Delta|Y)$ should provide better results than just $P(X|Y)$

- It is not clear to my why masking the anchor should provide better resutlts. Author say that masking "enforces an otherwise insuficient residual to make accurate prediction". If $\Delta$ is not sufficient for accurate prediction, shouldn't this lead to just overfitting the training set?

**Questions:**

- I don't understand if the anchor is chosen always from the training set, also during test. I think it should be, as the anchor label is used as ground truth. I think this should be made clearer

- How does RAM compare to sota methods for OOD or debiasing on tasks such as ImageNet-C? Does it achieve comparable performance or is there a significant gap?

---

> ### Author Response · Authors · 2023-11-18
> **Author Response - Reviewer BVnY**
>
> We thank you for the positive feedback. If our responses sufficiently address all the concerns raised, we request you to improve the score and champion our paper.
>
> **1. Why does Anchoring help?**
>
> Kindly check our detailed answer to this central question in the overall response. For completion, we include a summary here.
>
> In essence, anchoring involves representing an input $\mathrm{x}$ as a combination of an anchor $\mathrm{r}$ and the residual $\mathrm{x} - \mathrm{r}$, while maintaining the other training conditions of conventional neural networks. **This method capitalizes on the lack of shift invariance in Neural Tangent Kernels induced by common neural networks**. Anchored training employs different anchors for the same sample across epochs to marginalize the effect of anchor choice during inference. While it may seem akin to data augmentation, anchoring is fundamentally different in that it combinatorially expands the space of (anchor, residual) pairs with different anchor choices and enables exploration of a richer class of hypotheses, and does not manipulate the inputs. However, as our experiments show, **anchoring can be combined with any additional data augmentation technique**.
>
> **2. How does RAM work and Risk of Overfitting:**
>
> Kindly check our detailed answer to this central question in the overall response. For completion, we include a summary here.
>
> To clarify this, we begin by considering generalization to an OOD test sample. In the tuple $(\mathrm{r}, \Delta)$ for the test sample $\mathrm{x}_t$, the anchor $\mathrm{r} \in P(X)$ and it is possible that $\Delta = \mathrm{x}_t - \mathrm{r}$ can be novel to P($\Delta$). Thus, by exposing the model to more diverse combinations of $(\mathrm{r}, \Delta)$ during training, generalization can be improved. A naive way to improve the diversity of $P(\Delta)$ (or equivalently $P(X, \Delta)$) is to consider a wider anchor distribution (i.e., $P(R) \supset P(X) $). However, it is non-trivial to characterize the anchor distribution and arbitrarily improve its diversity; and increasing anchor diversity will lead to a combinatorially larger space (anchor, residual) pairs, thus requiring larger number of epochs to effectively converge.
>
> We adopt an alternative approach to increasing diversity of $P(\Delta)$ by **making the residuals noisy**, and propose a simple implementation in the form of RAM. Given the inherent challenge in defining a suitable residual noise distribution, we define the noise distribution to be same as the anchor distribution itself, i.e., $P(C) = P(N) = P(X)$, and implement it efficiently using the RAM regularizer. Formally, for a given tuple $(\mathrm{r},\mathrm{x} - \mathrm{r})$, anchor masking zeroes out the anchor while keeping the residual fixed, i.e., $(0, \mathrm{x} - \mathrm{r})$. Generally, the tuple for making a prediction for $\mathrm{x}$ with a zero anchor (note: zero vector is a valid anchor in our anchor distribution) will be written as $(0, \mathrm{x})$. However, **in anchor masking this can be interpreted as making a prediction using the zero anchor, but with a noised residual** $\mathrm{x} + \epsilon$ where $\epsilon = -\mathrm{r}$ and $\epsilon \in P(N)$.
>
> **3. Anchor Selection at Test Time:**
>
> During both training and testing, **the anchors are sampled from the anchor distribution $P(R)$**, which is set to the training distribution $P(X)$ itself. However, given that anchoring explicitly marginalizes the choice of anchor, **any random training sample (or a small number of them) can be used to obtain predictions at inference time**. We also emphasize that, during anchoring, the **anchor's label is never used** and the anchor sample only contributes to the residual computation.
>
> **4. Comparison with SoTA:**
>
> The key claim of this work is that, across dataset sizes (CIFAR-10 to Imagenet), architectures (ResNet to ViT) and network sizes (5M to 88M parameters), **anchored training can provide significant gains in OOD generalization, anomaly rejection and adaptation, compared to vanilla training**. We do not attempt to establish SoTA generalization performance, since top results are often obtained via model souping [1] or by fine-tuning large scale pre-trained models [2]. However, we emphasize that, **when the train recipe includes high-capacity architectures or advanced training strategies (Mixup, EMA, label-smoothing, cutmix), anchoring continues to improve over the base models** in all these cases, showing promise to be effective even with SoTA recipes involving large-scale pre-training and ensembling. In the updated conclusions, we have highlighted that integrating anchoring (w/ RAM) into these SoTA approaches is an important future direction of work.
>
> [1] M. Wortsman et al. Model soups: averaging weights of multiple fine-tuned models improves accuracy without increasing inference time, ICML 2022
>
> [2] S.Goyal et al. Finetune like you pretrain: Improved finetuning of zero-shot vision models, CVPR 2023.

---

> > ### Comment · Reviewer_BVnY · 2023-11-21
> >
> > Thanks for your response. I confirm my current score

---

> > > ### Author Response · Authors · 2023-11-21
> > > **Additional Comments**
> > >
> > > Thank you for your feedback.
> > > We wanted to check with you if you had any additional questions that we could answer before the discussion phase ends. Also, if you find our responses satisfactory, will you be open to raising the score to help champion our paper?

---

> > > > ### Comment · Reviewer_BVnY · 2023-11-21
> > > >
> > > > I do not have further questions. Regarding last point (OOD generalization), I find it reasonable that RAM does not beat sota OOD generalization techniques (but I do think that a comparison, perhaps in the appendix, can be relevant in the scope of the work). While I am not reasonably sure about raising to full accept, I will gladly raise my confidence score for my current rating.

---

### Author Response · Authors · 2023-11-18
**Overall Response**

We thank all the reviewers for their positive assessment of the paper. In addition to furnishing thorough responses to individual reviewer comments, we have also incorporated a consolidated response summarizing key points.

>  **Why does anchoring improve generalization?**

+ Simply put, anchoring involves transforming each input sample $\mathrm{x}$ into the tuple $(\mathrm{r}, \mathrm{x}-\mathrm{r})$, i.e., a combination of an anchor and the residual, while retaining all other aspects of a conventional neural network. Conceptually, as shown in [1], centering a dataset using different constant inputs will lead to different solutions, due to **inherent lack of shift invariance in neural tangent kernels** induced by commonly adopted neural networks. Anchored training uses different anchors for the same sample across different epochs with the **goal of marginalizing out the effect of anchor choice** at inference time. Furthermore, it implicitly enables the training process to explore a richer class of hypotheses.

+ While the idea of enforcing prediction consistency across different anchor choices might appear similar to data augmentation, we clarify that **anchoring does not impose any invariance to data characteristics**, but only expands the combination of (anchor, residual) pairs.

+ Regardless of the dataset size, architecture or size, anchoring produces improved OOD performance, while matching (or even surpassing) the ID performance, over standard training.

>  **How does RAM Improve Anchored Training?**

+  In the tuple $(\mathrm{r}, \Delta)$ for an OOD test sample $\mathrm{x}_t$, the anchor $\mathrm{r} \in P(X)$ and **it is possible that $\Delta = \mathrm{x}_t - \mathrm{r}$ can be novel to $P(\Delta)$**. Thus, by exposing the model to more diverse combinations of $(\mathrm{r}, \Delta)$ during training, generalization can be improved.

+ A naive way to improve the diversity of $P(\Delta)$ is to **consider a wider anchor distribution (i.e., $P(R) \supset P(X) $)**. However, this approach has two challenges: (a) It is non-trivial to characterize the anchor distribution (e.g., Imagenet data) and arbitrarily improve its diversity; (b) Even if we manage to do that, increasing anchor diversity will require larger number of epochs to effectively converge.

+ We adopt an alternative approach to increasing diversity of $P(\Delta)$ by **making the residuals noisy**, and propose an implementation in the form of RAM. Since defining a suitable residual noise distribution is challenging, we define the noise distribution to be same as the anchor distribution itself, i.e., $P(R) = P(N) = P(X)$. Formally, for a given tuple $(\mathrm{r}, \mathrm{x}-\mathrm{r})$, anchor masking zeroes out the anchor while keeping the residual fixed, i.e., $(0, \mathrm{x}-\mathrm{r})$. However, the tuple for making a prediction for $\mathrm{x}$ with a zero anchor (note: zero vector is a valid anchor) will be written as $(0, \mathrm{x})$. Hence, anchor masking can be interpreted as **making a prediction using the zero anchor, but with a noised residual** $\mathrm{x} + \epsilon$ where $\epsilon \in P(N)$.

+  **Network capacity plays a significant role in effectively leveraging the increased diversity** produced by RAM. For example, with ImageNet, as we move from RegNet (5M) to SWINv2-B (88M), we witness larger performance improvements over both anchoring w/o RAM as well as standard training (refer to updated Table 1 in the paper).

> **Choosing the Hyper-parameter $\alpha$**

-   The parameter **$\alpha$ controls the schedule of residual corruption** during training. Arbitrarily increasing $\alpha$ will start to adversely affect the training convergence and eventually the ID performance itself. Hence, **we recommend a nominal value of $\alpha=0.2$**, which we empirically find to converge in the same number of epochs as anchoring w/o RAM across all architectures, and to result in much improved generalization.

-   Similar to any existing approaches (e.g., Mixup) adopted for improving generalization, RAM with high $\alpha$ could have the risk of producing models with reduced sensitivity towards anomalous data. However, we find that with $\alpha = 0.2$, the **anomaly rejection trade-off is minimal** and interestingly, it can even lead to higher rejection AUROC scores using networks with higher capacity (e.g., SWINv2-B, updated Table 2 in the paper).

> **Summary of Changes made to the Paper Writing**

+   Reorganized the results section for improved clarity.
    * Additional architectures [DEIT-T, DEIT-S, SWINv2-T, SWINv2-T and SWINv2-B] and ImageNet OOD datasets (updated Table 1, 2)
    * Expanded adaptation results (updated Table 3, 4)
    * Robustness to label noise (Figure 5)

+   Expanded the background section to summarize some of the known results about anchoring.

+   Added a conclusion section.

[1] J.J. Thiagarajan et al. Single model uncertainty estimation via stochastic data centering, Neural Information Processing Systems, 2022.

---

### Author Response · Authors · 2023-11-21
**Request**

Dear reviewers,

As the author-reviewer discussion is nearing its final phase, we wanted to request you to check our responses and let us know if there are any additional comments or questions that we can address.

Here is a quick summary of our response:

+ We have clarified both through our response and edits to the paper draft **how anchoring helps in generalization** and why the proposed **RAM regularizer improves anchored training**.
+ Expanded our evaluation to include **multiple transformer architectures of varying complexity** (DEIT-T, DEIT-S, SWINv2-T, SWINv2-S, SWINv2-b and ViTb16) and **15 ImageNet OOD benchmarks**.
+ Our results show anchored training + RAM leads to **superior OOD generalization, calibration and anomaly rejection** across the board, where the benefits become more and more apparent with large-scale architectures.
+ Expanded our adaptation experiments (linear probing) with both **ID (multiple datasets including CIFAR-10 and CIFAR-100) and OOD evaluation (domainnet)** settings to demonstrate the quality of features from anchored training.
+ Added an additional experiment to illustrate the **robustness of anchored models to label noise** in training data.
+ Provided **memory overhead and training efficiency** comparison between standard training and anchored training. We find that the latter incurs 0.03 MB of additional memory and a minimal training overhead of 20 additional epochs.
+ Included a conclusion section with potential future directions of research.

We appreciate you for taking the time to review our work and providing useful feedback.

---

### Meta-Review · Area_Chair_U7jE · 2023-12-06

**Metareview:**

The paper presents Random Anchor Masking (RAM) in neural network training, enhancing model performance under domain shifts. This method involves applying an anchor (shifting the input) and masking the anchor with a probability. The empirical evaluation, conducted on diverse datasets like CIFAR-10/100 and ImageNet, demonstrates the effectiveness of RAM over standard anchoring techniques.

Strengths
- RAM is simple to integrate into existing models, only affecting the input layer.
- The methodology has been thoroughly tested across multiple datasets and architectures.
- The paper is well-structured and clearly written.

Weaknesses
- The paper lacks either intuition or a theoretical underpinning on why the strategy would work. Similarly, it doesn't provide much ablations or variants of methods to better understand when and how the strategy improves robustness. This was a contention raised universally by all reviewers, and one I strongly agree with.
- Baselines are lacking, comparing only to vanilla models and anchored models. There is a wide variety of robustness techniques that is not included here that I worry the gains compared to the baselines are overstated. For example, why not compare to the many data augmentation strategies that similarly increase robustness based on modifying the inputs? There is some discussion of the differences between anchoring and data augmentation, and this would be better substantiated with experiments. The lack of baselines contributes to the lack of understanding behind the method, and I believe this particular weakness was missed by certain reviewers.

This is a somewhat borderline paper with no strong evidence to weigh in favor of it. Overall, I'm not convinced the work is a valuable contribution in terms of learnings or as a method that should indeed be generally adopted. Specifically, it is not clear when and how this method would improve generalization, which I would recommend the authors perform more careful studies to understand. In addition, I recommend the authors perform more comprehensive experiments on baselines to at least obtain empirical intuitions in lieu of technical ones.

**Justification For Why Not Higher Score:**

See weaknesses above.

**Justification For Why Not Lower Score:**

N/A

---

### Decision · Program_Chairs · 2024-01-16

Reject